# Revisiting Differential Attention: A Fine-Tuning Perspective on Practical Noise Mitigation

## Abstract

The self-attention mechanism in Transformer models is widely adopted but remains vulnerable to attention noise. Differential Transformer and its variant DEX attempt to address this issue; however, the former requires training from scratch, while the latter cannot directly mitigate noise during the attention computation process. In this paper, we propose DAA (Differential Attention Adaption), a novel method that can both reduce attention noise and be flexibly inserted during the fine-tuning stage. Specifically, DAA introduces lightweight learnable modules in the process of calculating attention scores, implementing the differential mechanism to suppress noise. We find that DAA can offset attention noise while introducing few parameters (less than 1% of the total model parameters) and directly act on the updates of the K and Q matrices, achieving effects similar to those of a Differential Transformer model trained from scratch. We further compare our approach with two methods that explore different positions of differentiation: one modifies the input sequence to separately compute K, Q, or V, while the other regulates the output matrix (DEX). Experimental results show that DAA can better effectively improve model performance with a small amount of fine-tuning data.

## 1 Introduction

The Transformer architecture has become the cornerstone of modern language models and a pivotal technology in a wide array of artificial intelligence applications Vaswani et al. (2017); Dosovitskiy et al. (2020); Radford et al. (2021); Kirillov et al. (2023); Carion et al. (2020). Although Transformers are widely used, many studies have shown that this architecture has difficulties in retrieving key information due to the presence of attention noise.This misallocation of focus can degrade language model performance, particularly in tasks involving long sequences or complex data Liu et al. (2023); Lu et al. (2021).

In response to this critical issue, researchers have proposed Differential Transformer Ye et al. (2025), a novel architecture to reduce attention noise, inspired by differential amplifiers in electrical engineering. It computes the difference between two parallel softmax attention maps to suppress noise and amplify the signal from relevant tokens. While effective, the Differential Transformer necessitates training a model from scratch, preventing its application to the existing pretrained language models. To circumvent the need for complete retraining, some other methods are introduced Wu et al. (2025); Kong et al. (2025). For example, OpAmp adaption shows excellent results in the fine-tuning process, but complex processing of fine-tuned data is required in advance Wu et al. (2025). The other method, DEX (Differential Extension), aims to integrate the benefits of Differential Transformer into the normal fine-tuning stage by applying a learnable differential operation to the output value matrix Kong et al. (2025). However, it does not directly intervene in the attention score calculation, thus failing to mitigate noise during the crucial attention computation phase.

Building on these insights, we introduce Differential Attention Adaption (DAA), a novel method designed to overcome the limitations of both the Differential Transformer and DEX when applied to conventional training datasets. Our approach inserts learnable modules directly into the self-attention mechanism, implementing a differential operation during the calculation of attention scores. Specifically, these modules act on the product of the Key (K) and Query (Q) matrices, allowing DAA

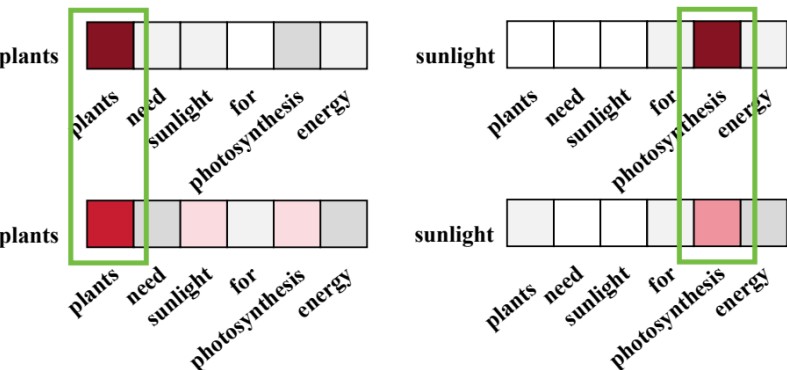

Figure 1: Attention scores on ARC-E (Science Question Answering Generation task) for DAA (top) and DEX (bottom). Darker red indicates stronger attention. Green boxes highlight that DAA demonstrates more focused and accurate attention on core scientific associations (e.g., plants→plants subject continuity, sunlight→photosynthesis key scientific logic) compared to DEX.

Table 1: Comparison of differential attention architectures, where $d_{\text{model}}$ represents the dimension of the model's hidden states.

| Architecture | Reduce Attention Noise | Introduced to Existing Transformer Models | # Parameters of Each Attention Layer ($h$ heads) |
|---|---|---|---|
| DIFF Transformer | ✓ | ✗ | $7d_{\text{model}}^2$ |
| DEX | ✗ | ✓ | $(4 + \frac{1}{h})d_{\text{model}}^2$ |
| DiffK (Ours) | ✓ | ✓ | $5d_{\text{model}}^2$ |
| DiffQ (Ours) | ✓ | ✓ | $5d_{\text{model}}^2$ |
| DiffV (Ours) | ✗ | ✓ | $5d_{\text{model}}^2$ |
| DAA (Ours) | ✓ | ✓ | $(4 + \frac{1}{h})d_{\text{model}}^2$ |

to mitigate attention noise at its source. This direct intervention achieves an effect analogous to a Differential Transformer but without the need for training the model from scratch. To validate our approach, we compare DAA against two alternative differentiation strategies: one that alters the input sequence to compute K, Q, or V separately, and another (DEX) that adjusts the final output matrix. Our analysis reveals that, unlike these methods, which tend to focus excessively on either local or global features, DAA integrates a differential mechanism throughout the entire attention score computation process. This holistic approach enables DAA to effectively offset attention noise with the significant advantage of being applicable to pre-trained models.

As highlighted in Table 1, DAA uniquely combines the key advantages of its predecessors. It effectively reduces attention noise, similar to the original Differential Transformer, but crucially, it can be applied to existing pre-trained models. Furthermore, it achieves this with the same parameter efficiency as DEX, introducing a minimal number of new parameters (less than 1% of the total model parameters). As experimental results demonstrate, DAA enhances model performance with only a small amount of fine-tuning data effectively, offering a practical and efficient solution for fine-tuning pretrained Transformer models.

## 2 BACKGROUND

The attention noise in Transformer models has spurred the development of novel architectures aimed at enhancing signal clarity during the self-attention process. In this section, we review two significant preceding works : the Differential Transformer and its lightweight adaptation, DEX.

## 2.1 DIFFERENTIAL TRANSFORMER

Inspired by differential amplifiers in electrical engineering, the Differential Transformer Ye et al. (2025) introduces a novel attention mechanism, known as DIFF attention, to actively suppress attention noise and amplify relevant signals within the input sequence. This is achieved by computing the difference between two parallel attention maps, which effectively cancels out common-mode noise.

The core of the Differential Transformer lies in its unique formulation of the attention mechanism. Given an input sequence $X \in \mathbb{R}^{N \times d_{\text{model}}}$, it first generates two distinct sets of queries $(Q_1, Q_2)$ and keys $(K_1, K_2)$ from separate learnable projection matrices, while sharing a single value matrix $V$. The differential attention is then computed as follows:

$$[Q_1; Q_2] = XW_Q, \quad [K_1; K_2] = XW_K, \quad V = XW_V,$$
$$A_1 = \text{softmax}\left(\frac{Q_1 K_1^T}{\sqrt{d}}\right),$$
$$A_2 = \text{softmax}\left(\frac{Q_2 K_2^T}{\sqrt{d}}\right), \quad (1)$$
$$O' = (A_1 - \lambda A_2)V,$$

where $W_Q, W_K, W_V \in \mathbb{R}^{d_{\text{model}} \times 2d}$ are learnable parameter matrices, $Q_1, Q_2, K_1, K_2 \in \mathbb{R}^{N \times d}$ and $V \in \mathbb{R}^{N \times 2d}$ denote projected matrices. $A_1, A_2$ are the softmax attention scores, $\lambda$ is a learnable scalar that balances the contribution of the two attention maps. $O'$ is the differential attention output.

The primary advantage of the Differential Transformer is its remarkable effectiveness in reducing attention noise, leading to sparser and more focused attention patterns. This noise cancellation enhances the model's ability to identify and prioritize key information, which has been shown to improve performance on a variety of downstream tasks. By design, it directly intervenes in the attention score calculation to improve the signal-to-noise ratio. However, despite its innovative approach, the Differential Transformer has a significant limitation: it requires training a model from scratch. This necessity prevents its direct application to the vast number of powerful, pre-existing language models.

## 2.2 DEX (DIFFERENTIAL EXTENSION)

To address the training-from-scratch limitation of the Differential Transformer, DEX (Differential Extension) Kong et al. (2025) is proposed as a more lightweight and flexible alternative. DEX is designed to integrate the principles of differential mechanisms into pre-trained models without requiring complete retraining.

Instead of modifying the core attention score computation, DEX applies a differential adaptation to the output of the attention heads. The standard attention scores $A$ are first calculated. Then, a differential update is applied:

$$Q = XW_Q, \quad K = XW_K, \quad V = XW_V$$
$$A = \text{softmax}\left(\frac{QK^T}{\sqrt{d}}\right) \quad (2)$$
$$O' = AV(I - \lambda(t))W_{\text{DEX}},$$

where $W_{\text{DEX}}$ is a learnable matrix, initialized as an identity matrix, $I$ is an identity matrix. $\lambda(t)$ is a time-dependent weighting factor. This approach allows for targeted updates (attention layer parameters) while keeping most of the model parameters, including the feed-forward networks, frozen during training.

The main strength of DEX lies in its high adaptability and efficiency. It can be integrated into existing pre-trained models, avoiding the prohibitive costs associated with training a large model from the ground up. By focusing the updates on a small subset of parameters within the self-attention module, it provides a lightweight solution for fine-tuning.

However, since the differential operation is applied to the output value matrix after the attention scores have been computed, DEX does not directly address the attention noise during the process of

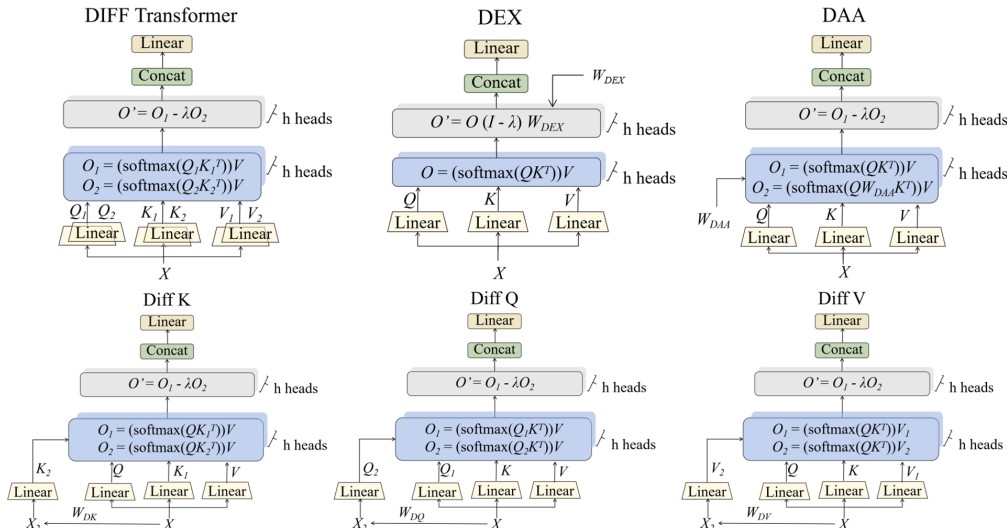

Figure 2: Comparison of DEX, DAA, and Differentiate $K, Q$, or $V$ to Differential Attention archi-
tectures. DEX inserts a learnable matrix to regulate output during the attention output stage; DAA
inserts a learnable matrix to regulate $Q - K$ connections during the attention score calculation stage;
the last three methods transform the input hidden states $X$ when they are projected into $Q, K$, or $V$
matrices, to differentiate attention output.

score calculation. This is inconsistent with the Differential Transformer, which eliminates attention
noise directly during computation.

While both the Differential Transformer and DEX represent significant advancements in mitigating
attention noise, they possess notable limitations. The former requires complete model retraining,
and the latter only indirectly addresses noise in attention scores. This creates a clear need for a
method that is not only highly adaptable to pre-trained models but also directly optimizes attention
computations. Our proposed method is designed to fill this critical gap.

## 3 METHOD

Based on the analysis of Differential Transformer and DEX in the last Section, we discuss methods
for adjusting the attention mechanism during the fine-tuning phase to reduce attention noise in this
section. Specifically, our core idea is to introduce a lightweight, learnable module similar to DEX
and combine it with the existing self-attention mechanism to achieve an explicit differential attention
mechanism. Depending on the position where the module is inserted, we explore two approaches
to perform differential computation: (1) directly inserting it into the attention score calculation (our
proposal DAA), and (2) inserting it at the input sequence level, before projecting onto the query, key,
or value matrices. In the following subsections, we will detail the components of our framework and
their theoretical foundations.

### 3.1 WHY DIFFERENTIAL ADAPTION WORKS

Standard Transformer models (such as Llama) are highly correlated with differential Transformer
models in the absolute values of attention scores Kong et al. (2025), indicating that both Atten-
tions consistently identify important information. Differential Transformer enhances flexibility by
introducing negative attention scores Lv et al. (2024), enabling better differentiation of noise infor-
mation. These factors allow the self-attention of pretrained models to transition appropriately to the
differential attention.

Both the Differential Transformer (equation 4) and DEX (equation 2) introduce $\lambda$ to regulate the dif-
ferential magnitude. In the Differential Transformer, $\lambda$ is reparameterized through several learnable

vectors rather than being learned as a single scalar, which helps improve its learning stability and expressive power. In DEX, $\lambda$ introduces an annealing mechanism. During the early stages of fine tuning, $\lambda$ gradually increases from an initialized zero value, guiding the model to adopt the differential mechanism; in the later stages of fine tuning, $\lambda$'s value is fully learned by the model, allowing it to adaptively adjust the differential strength. In this paper, we adopt the $\lambda$ mechanism from DEX to regulate the differential mechanism, enabling the model to maximize the inheritance of pre-trained knowledge while preliminarily introducing the differential mechanism to enhance performance.

## 3.2 DIFFERENTIAL ARCHITECTURES VIA $Q - K$ INTERACTION (DAA)

Our proposed method, DAA (Differential Attention Adaption), directly regulates the attention score calculation process to reduce attention noise. Instead of creating parallel attention mechanisms, DAA introduces a learnable matrix into the query-key interaction. Specifically, for each attention head, we introduce a small, learnable differential matrix $W_{\text{DAA}} \in \mathbb{R}^{d_k \times d_k}$, which is initialized as an identity matrix to preserve the pretrained knowledge at the beginning of fine-tuning.

The DAA mechanism computes two attention score matrices. The standard attention scores $A_1$ are also first calculated, $A_2$ and then computed by applying the differential matrix $W_{\text{DAA}}$ to the dot product of the query matrix $Q$ with the key matrix $K$. The final attention distribution is the difference between these two scores, modulated by a dynamic weight $\lambda(t)$ like DEX Kong et al. (2025):

$$\lambda(t) = (1 - \alpha)\left[\frac{t}{T}\lambda_{\text{init}}\right] + \alpha\lambda_{\text{learn}}, \quad \alpha = \min\left(1, \frac{t}{T}\right) \tag{3}$$

where $t$ is the current step in training, $T$ is the annealing duration, $\lambda_{\text{learn}}$ is a learnable parameter initialized around zero, and $\lambda_{\text{init}}$ is a constant.

The computation of attention output for a single head $i$ is as follows:

$$A_1 = \text{softmax}\left(\frac{QK^T}{\sqrt{d}}\right),$$
$$A_2 = \text{softmax}\left(\frac{QW_{\text{DAA}}K^T}{\sqrt{d}}\right) \tag{4}$$
$$O' = (A_1 - \lambda(t)A_2)V$$

This formulation allows the model to learn to subtract a noise pattern identified by the second attention scores, directly improving the final attention distribution. Because $W_{\text{DAA}}$ has only $d_k \times d_k$ parameters in each head, this adaptation is extremely lightweight.

## 3.3 DIFFERENTIAL ARCHITECTURES VIA INPUT DIFFERENTIATION

We also explore alternative methods, transforming the input hidden states $X \in \mathbb{R}^{N \times d_{\text{model}}}$ when they are projected into $Q, K$, or $V$ matrices, to implement differential attention. These methods all introduce a single learnable identity matrix $W_D \in \mathbb{R}^{d_{\text{model}} \times d_{\text{model}}}$, ensuring a stable start to training by beginning with a standard attention configuration. The core idea is to generate a differentiated representation of the input and use the difference between the standard and differentiated pathways to calculate the attention output.

**Differential Adaption via Query.** This architecture generates a different query matrix, $Q_2$, from the transformed input sequence $X' = XW_{D_Q}$. The standard query, $Q_1$, is computed from the original input $X$. The attention is then calculated as the difference between two attention maps. The subtraction of attention maps derived from a primary query ($Q_1$) and a differentiated query ($Q_2$) acts as a differential mechanism:

$$Q_1 = XW_Q, \quad Q_2 = X'W_Q = (XW_{D_Q})W_Q$$
$$A_1 = \text{softmax}\left(\frac{Q_1 K^T}{\sqrt{d_k}}\right)$$
$$A_2 = \text{softmax}\left(\frac{Q_2 K^T}{\sqrt{d_k}}\right) \tag{5}$$
$$O' = (A_1 - \lambda(t)A_2)V$$

**Differential Adaption via Key.** Analogously, we can calculate another key matrix, $K_2$, generated from the transformed input $X'$. While the standard key, $K_1$, is derived from $X$. This approach evaluates the query against two different content representations (via $K_1$ and $K_2$):

$$K_1 = XW_K, \quad K_2 = X'W_K = (XW_{D_K})W_K$$
$$A_1 = \text{softmax}\left(\frac{QK_1^T}{\sqrt{d_k}}\right)$$
$$A_2 = \text{softmax}\left(\frac{QK_2^T}{\sqrt{d_k}}\right) \tag{6}$$
$$O' = (A_1 - \lambda(t)A_2)V$$

**Differential Adaption via Value.** This approach modifies the value stream directly. Instead of subtracting attention distributions, it computes a modified value matrix $V'$ by applying a dynamically weighted differential transformation to the input sequence. This is distinct from DEX, as the modification occurs before the final attention-weighted sum. The method effectively creates a primary value stream ($V_1$) and a secondary stream ($V_2$) that is subtracted from it. This can be viewed as a learned, dynamic feature suppression mechanism that filters irrelevant information from the retrieved content itself, rather than altering the attention scores.

$$W_E = I - \lambda(t)W_{D_E}$$
$$V' = (XW_E)W_V = XW_V - \lambda(t)XW_{D_E}W_V = V_1 - \lambda(t)V_2$$
$$A = \text{softmax}\left(\frac{QK^T}{\sqrt{d_k}}\right) \tag{7}$$
$$O' = AV' = AV_1 - \lambda(t)AV_2$$

Here, $W_E$ serves as an effective transformation matrix that directly modulates the information carried by the value vectors.

## 3.4 THEORETICAL ANALYSIS OF ADAPTATION STRATEGIES

Among the various differential fine-tuning adaptations, DAA is theoretically positioned as the most effective due to its direct and holistic intervention in the attention score calculation process. The principal source of attention noise is often the computation of inaccurate similarity scores within the $QK^T$ dot product, which can arise from spurious correlations or "common-mode" distractions where irrelevant tokens receive undue attention Ye et al. (2025).

We can formally model the computed attention logits, $S_{\text{computed}}$, as the sum of an ideal, noise-free signal, $S_{\text{ideal}}$, and a noise component, $\xi$:

$$S_{\text{computed}} = S_{\text{ideal}} + \xi = \frac{QK^T}{\sqrt{d_k}} + \xi \tag{8}$$

The goal of a noise mitigation strategy is to suppress the influence of the noise matrix $\xi$ before the softmax function, which can otherwise amplify these erroneous signals and degrade model performance Liu et al. (2023). The attention noise $\xi$ is sampled from a multivariate normal distribution, $\xi \sim \mathcal{N}(0, \sigma_p^2 I_d)$. The symbol $\mathcal{N}$ denotes a multivariate normal distribution. The parameter, $\sigma_p^2 I_d$,

is the covariance matrix. Here, $\sigma_p^2$ (sigma-p squared) is the variance, which measures the spread or power of the noise. $\boldsymbol{I}_d$ is the d-dimensional identity matrix.

DAA addresses this challenge directly. By generating a primary attention map from the noisy logits and subtracting a secondary, corrective map, it performs an explicit noise cancellation operation. The two attention distributions are:

$$A_1 = \text{softmax}(S_{\text{computed}}) \quad \text{and} \quad A_2 = \text{softmax}\left(\frac{QW_{\text{DAA}}K^T}{\sqrt{d_k}}\right) \tag{9}$$

The core hypothesis is that the lightweight, learnable matrix $W_{\text{DAA}}$ enables the model to learn a transformation that isolates the noise pattern. During fine-tuning, the model is incentivized to learn a $W_{\text{DAA}}$ such that the secondary logits approximate the noise component itself:

$$A_1 - \lambda A_2 \approx \text{softmax}(S_{\text{ideal}}) \tag{10}$$

By subtracting the resulting attention map, $O' = (A_1 - \lambda A_2)V$, DAA directly counteracts the noise within the attention distribution. This mechanism is a close parallel to the common-mode signal rejection found in differential amplifiers, which is the original inspiration for the Differential Transformer Ye et al. (2025).

In contrast, other architectures offer more indirect solutions. Input differentiation methods (DiffQ, DiffK) alter one of the core components of the attention calculation. For example, DiffQ computes its secondary attention map using a transformed query, $Q' = (XW_{D_Q})W_Q$. While this generates a different attention map, it is less direct because the noise $N$ arises from the interaction of the *original* $Q$ and $K$. The model must learn a global transformation $W_{D_Q}$ on the entire hidden state in the hope that the resulting $Q'$ will produce an attention map suitable for subtraction, rather than directly modeling the noisy interaction itself.

Similarly, post-hoc correction methods like DEX operate after the potentially noisy attention scores have already been computed and applied. The DEX operation is applied to the final output:

$$O' = (A_1V)(I - \lambda(t)W_{\text{DEX}}) \tag{11}$$

Here, the attention map $A_1 = \text{softmax}(S_{\text{computed}})$ is already corrupted by $N$. DEX can only attempt to filter the output by transforming the weighted value vectors; it cannot rectify the misallocated attention weights within $A_1$. DAA, by intervening at the critical stage of score calculation, provides a more principled and direct mechanism for noise mitigation, which we expect to yield superior performance.

## 4 Experiments and Analysis

We first conduct comparative experiments on different differential adaptation methods using the pre-trained language model GPT-2 (117M) Radford et al. (2019). Subsequently, we introduce the DAA architecture into the Llama-3.2-1B and Llama-3.1-8B models Dubey et al. (2024) for fine-tuning to validate the generality of the DAA architecture. The comparative experiments quantitatively validate the effectiveness of DAA in eliminating attention noise and improving model performance.

### 4.1 Differential Adaption for Foundational Language Modeling

**Experimental Settings.** We introduce five differential attention architectures (DEX; DAA; differentiate via key, query, and value) into the pre-trained GPT-2 model to construct five new models. To ensure that the fine-tuned models still keep basic capabilities, we select a subset of the OpenWeb-Text dataset (OWT) Peterson et al. (2019) as the fine-tuning data source (this dataset is similar to the GPT-2 pre-training data). Finally, we fine-tune the five new architecture models and the standard attention mechanism (eager) GPT-2 model on the fine-tuning dataset.

**Experimental Results and Analysis.** During training, every model records the training loss every 100 steps, and the results are shown in Figure 3.

Table 2: Comparison of models before and after fine-tuning with different attention architectures. FT represents the standard fine-tuning method. PPL stands for Perplexity, lower is better. ACC stands for Accuracy, higher is better. Bold values indicate the best performance in that column. WKT2 represents the WikiText2 corpus, NR represents the needle retrieval.

| Model | OWT (PPL) | OWT(new) (PPL) | WKT2 (PPL) | LAMBADA (PPL) | LAMBADA (ACC) | NR (ACC) |
|---|---|---|---|---|---|---|
| *GPT-2 (117M)* | | | | | | |
| Base | 33.3 | 32.4 | 24.7 | 42.98 | 47.3 | 84.9 |
| + FT | 29.0 | **28.1** | 25.6 | 43.1 | 48.1 | 85.3 |
| + DEX | **28.9** | 28.2 | 25.2 | 42.95 | 48.2 | 87.4 |
| + DiffQ (Ours) | 29.2 | 28.4 | **24.4** | 43.55 | 48.0 | 84.4 |
| + DiffK (Ours) | 29.3 | 28.5 | **24.4** | 43.55 | 48.1 | 88.3 |
| + DiffV (Ours) | 29.1 | 28.3 | 25.3 | 43.21 | 48.1 | 84.4 |
| + DAA (Ours) | **28.9** | **28.1** | 24.7 | **42.25** | **48.9** | **89.5** |

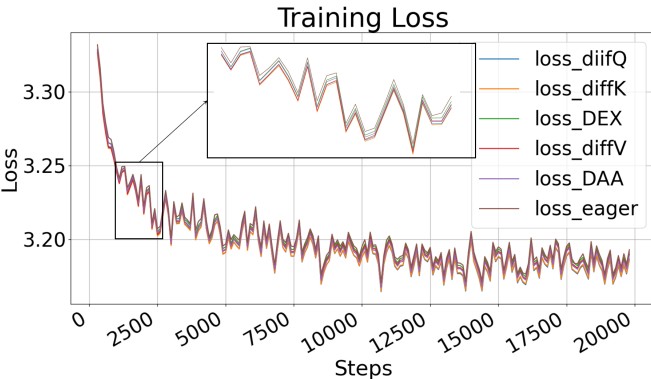

Figure 3: Training loss of different attention architectures

According to the experimental results, the two architectures, differential adaption via key and query, keep the low loss throughout the entire training process. The training loss of DAA is lower than DEX, while the training loss of the standard attention mechanism is the highest. According to Liu et al. (2020), training loss of the model in the general dataset does not fully reflect its generalization performance. In order to test the general capabilities of the models, we also conduct a performance analysis of these models, and the results are shown in Table 2.

The evaluation results reveal the distinct advantages of the DAA architecture. While all fine-tuning methods demonstrate improved perplexity on the OpenWebText dataset compared to the base GPT-2 model, their performance diverges on downstream tasks that are more sensitive to attention quality. Most notably, DAA achieves the highest accuracy on both the LAMBADA Grave et al. (2016) and Needlehaystack benchmarks. This strongly indicates its superior ability to mitigate attention noise and focus on relevant tokens. While DiffQ and DiffK show competitive perplexity on WikiText2, they do not match DAA's gains in the more challenging retrieval and reasoning tasks. The performance of DiffV, which is comparable to standard fine-tuning, suggests that altering the value stream is less effective than directly intervening in the attention score computation. Collectively, these results validate our hypothesis that directly modulating the Q-K interaction, as DAA does, provides a more effective and robust mechanism for improving model performance by reducing attention noise during fine-tuning.

## 4.2 DIFFERENTIAL ADAPTION FOR MULTI-TASKS MODELING

**Experimental Settings.** We further apply the five differential attention architectures (DEX; DAA; differentiate via key, query, value) to the Llama-3.2-1B, Llama-3.1-8B models. Since the models all have undergone pre-training, we select a subset of the allenai/tulu-3-sft-olmo-2-mixture-0225 dataset as fine-tuning data. The allenai/tulu-3-sft-olmo-2-mixture-0225 dataset is a large-scale, multilingual text dataset released by Allen Institute for Artificial Intelligence (AllenAI), specifically designed for supervised fine-tuning (SFT) of language models OLMo et al. (2024). This corpus con-

Table 3: Green indicates improvement over the baseline, while gray indicates a decrease.

| Model | Arc-E | Arc-C | BoolQ | Hellaswag | OBQA | WIC | Winogrande | WSC | AVG | Δ |
|---|---|---|---|---|---|---|---|---|---|---|
| *Llama-3.2-1B* | | | | | | | | | | |
| Base | 62.08 | 36.3 | 61.8 | 63.6 | 34.6 | 48.6 | 56.49 | 36.5 | 51.31 | - |
| + FT | 65.08 | 33.56 | 57 | 64.13 | 30.8 | 50.1 | 56.2 | 39.42 | 49.54 | -1.77 |
| + DEX | 65.86 | 34.92 | 60.24 | 64.16 | 31.6 | 48.43 | 56.75 | 53.65 | 51.93 | +0.62 |
| + DiffQ (Ours) | 65.26 | 35.25 | 58.97 | 64.06 | 31.6 | 48.8 | 54.2 | 55.29 | 51.68 | +0.37 |
| + DiffK (Ours) | 66.31 | 35.59 | 62.51 | 64.11 | 29.4 | 48.8 | 49.3 | 55.72 | 50.1 | +0.37 |
| + DiffV (Ours) | 67.72 | 31.86 | 57.61 | 64.03 | 28.4 | 50.2 | 56.91 | 43.27 | 50.0 | -1.31 |
| + DAA (Ours) | 65.96 | 33.22 | 62.78 | 64.29 | 33.4 | 49.9 | 56.51 | 52.88 | 52.37 | +1.06 |
| *Llama-3.1-8B* | | | | | | | | | | |
| Base | 78.9 | 52.6 | 74.9 | 78 | 42.1 | 51.9 | 73.1 | 58.6 | 63.76 | - |
| + FT | 76.01 | 52.2 | 74.39 | 80.5 | 43.2 | 52.8 | 73.46 | 58.94 | 63.94 | +0.18 |
| + DEX | 77.2 | 52.66 | 78.4 | 78.6 | 42.3 | 52.5 | 73.9 | 59.1 | 64.33 | +0.57 |
| + DiffQ (Ours) | 78.6 | 51.39 | 73.21 | 79.3 | 43.8 | 52 | 70.62 | 56.54 | 63.18 | -0.58 |
| + DiffK (Ours) | 75.49 | 51.5 | 74.83 | 78.6 | 44.4 | 52.2 | 70.54 | 58.65 | 63.28 | -0.48 |
| + DiffV (Ours) | 76.72 | 50.51 | 78.52 | 78.5 | 44.4 | 52.1 | 71.8 | 53.51 | 63.26 | -0.5 |
| + DAA (Ours) | 76.9 | 52.71 | 79.04 | 77.4 | 42.8 | 53.3 | 74.25 | 59.62 | 64.50 | +0.74 |

tains 552M tokens (Llama-3 tokenizer), significantly smaller than the dataset size used for models pretraining.

**Experimental Results and Analysis.** We report performances on 8 widely used language modeling benchmarksClark et al. (2018); Wang et al. (2019); Mihaylov et al. (2018); Bisk et al. (2020); Sakaguchi et al. (2021). The experimental results, summarized in Table 3, unequivocally establish the superior performance of our proposed DAA architecture across different model scales in the multi-task fine-tuning context. A crucial initial observation is the suboptimal performance of standard fine-tuning (FT). For the Llama-3.2-1B model, standard FT leads to performance degradation relative to the base model. While the larger Llama-3.1-8B model does not degrade, it sees only a negligible gain. This highlights a key challenge: standard fine-tuning with limited data can harm or fail to improve a model's general capabilities, likely due to catastrophic forgetting or overfitting Kirkpatrick et al. (2017).

In contrast, the various differential adaptation methods show divergent outcomes, revealing the importance of where the differential mechanism is applied. The architectures that differentiate the input sequence (DiffQ, DiffK, and DiffV) produce inconsistent and ultimately poor results. While DiffQ and DiffK offer marginal gains on the 1B model, they are detrimental to the performance of the 8B model, causing average scores to drop. The DiffV method is the least effective, resulting in a performance decrease for both the 1B and 8B models. This strongly suggests that modifying the value stream after attention scores are computed, or altering the input streams in isolation, is a less robust strategy for noise mitigation.

The DAA and DEX methods, however, consistently improve upon the base models. While the existing lightweight method, DEX, provides a solid improvement and successfully counteracts the degradation seen in standard FT, our proposed DAA method achieves the most substantial and consistent performance gains across both model scales. By directly intervening at the core of the attention score computation—the Q-K interaction—DAA is able to more effectively model and subtract attention noise at its source (as shown in Figure 1). Unlike methods that apply localized changes or post-hoc corrections, DAA's holistic modulation of the query-key relationship proves to be a more principled and impactful mechanism for enhancing model performance during fine-tuning.

## 4.3 ANALYSIS OF DAA MECHANISM

To provide an intuitive understanding of how DAA mitigates attention noise and enhances the signal-to-noise ratio, we conduct a visualization analysis of the attention patterns. We compare the fine-tuned Standard GPT-2 model with the DAA-enhanced GPT-2 model on a specific case, visualizing the distribution of attention scores and their underlying components. The results are presented in Figure 4.

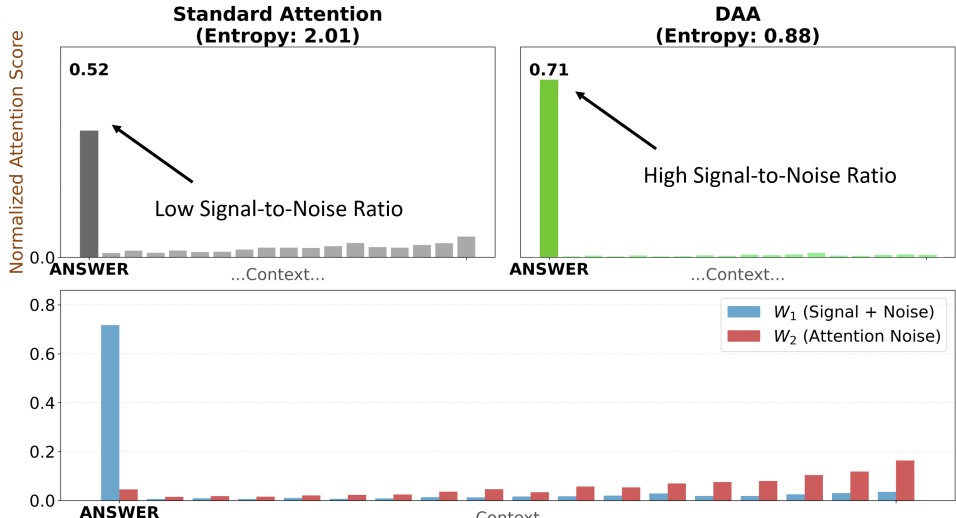

Figure 4: Visualization of attention mechanisms on the GPT-2 model. **Top Row:** Comparison of normalized attention scores between Standard Attention (left) and DAA (right). The entropy values indicate the concentration of attention. **Bottom Row:** Decomposition of DAA's attention composition, showing the primary attention map ($W_1$, effectively $A_1$) and the learned noise map ($W_2$, effectively $A_2$). The subtraction mechanism ($W_1 - \lambda W_2$) effectively filters out background noise.

**Entropy and Focus.** As shown in the top row of Figure 4, the Standard Attention mechanism exhibits a relatively dispersed attention distribution with a high entropy. The model assigns significant weight to the ANSWER token but also wastes considerable attention mass on irrelevant context tokens, indicating a low signal-to-noise ratio. In contrast, the DAA mechanism produces a much sharper attention distribution with a significantly lower entropy. The demonstrates DAA's ability to concentrate focus on critical information while suppressing distractions.

**Noise Cancellation Mechanism.** The bottom row of Figure 4 reveals the inner workings of the DAA module. It can be observed that $W_2$ effectively captures the background noise distribution across the context tokens, while keeping the value on the ANSWER token extremely low. By performing the differential operation (approximately $W_1 - W_2$), the model successfully cancels out the noise on the context tokens. This confirms that DAA does not merely reshape the output arbitrarily; instead, it explicitly models and subtracts the noise component during the attention score calculation, thereby recovering a cleaner signal for subsequent layers.

## 5 CONCLUSION

In this work, we propose Differential Attention Adaption (DAA), a parameter-efficient fine-tuning method that directly mitigates attention noise by inserting a learnable module into the core query-key computation. Unlike methods that require training from scratch or apply corrections after attention outputs, DAA intervenes at the source of noise generation. Our experiments on several language models confirm that DAA significantly outperforms standard fine-tuning and other adaptive differential techniques, successfully enhancing performance without causing catastrophic forgetting. However, a limitation of our approach is the requirement to compute a secondary attention map, which introduces a marginal computational overhead during inference compared to standard attention. Future work may focus on optimizing this efficiency and extending DAA to multi-modal tasks. Overall, DAA presents a principled, practical solution for fine-tuning pre-trained Transformers.

## 6 REPRODUCIBILITY STATEMENT

We have taken the necessary steps to ensure the reproducibility of our results. Specifically, Section 4.1 discusses the general experiment settings in our paper. Appendix B provides the detailed steps to collect and process the datasets used in downstream tasks. Appendix D includes the detailed steps to construct the fictitious synthetic data used by our method. Finally, Appendix E and the supplementary material list the implementation details of our method and all baselines, including the codebase, training hyperparameters, evaluation details, etc.

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

## A    USE OF LARGE LANGUAGE MODELS

During the preparation of this work, the authors used Large Language Models (LLMs) to assist with editing and refining the language. The LLMs were primarily used for improving grammar, clarity, and phrasing of the manuscript. All scientific contributions, including the core ideas, experimental design, and analysis of results, were conceived and executed by the human authors.

## B    # PARAMETERS OF EACH ARCHITECTURE'S ATTENTION LAYER

In this section, we provide a detailed derivation of the number of parameters for each attention layer in the architectures discussed in this paper and compared in Table 1. We use the following notation:

- $d_{\text{model}}$: The dimension of the model's hidden states.
- $h$: The number of attention heads.
- $d_k$: The dimension of the key and query vectors for each head, where $d_k = d_{\text{model}}/h$.
- $d_v$: The dimension of the value vectors for each head, where $d_v = d_{\text{model}}/h$.
- $N$: sequence length.

For simplicity and consistency with the standard Transformer architecture, we assume $d_k = d_v$.

### B.1    STANDARD TRANSFORMER ATTENTION

A standard multi-head attention layer consists of four main learnable weight matrices:

1. **Query projection** ($W_Q$)**:** Maps the input hidden states to the query space. Dimensions: $d_{\text{model}} \times d_{\text{model}}$.

2. **Key projection** ($W_K$)**:** Maps the input hidden states to the key space. Dimensions: $d_{\text{model}} \times d_{\text{model}}$.

3. **Value projection** ($W_V$)**:** Maps the input hidden states to the value space. Dimensions: $d_{\text{model}} \times d_{\text{model}}$.

4. **Output projection** ($W_O$)**:** Maps the concatenated output of the attention heads back to the hidden state dimension. Dimensions: $d_{\text{model}} \times d_{\text{model}}$.

The total number of parameters is the sum of the parameters of these four matrices:

$$N_{\text{Standard}} = d_{\text{model}}^2 + d_{\text{model}}^2 + d_{\text{model}}^2 + d_{\text{model}}^2 = 4d_{\text{model}}^2$$

### B.2    DIFFERENTIAL (DIFF) TRANSFORMER

The Differential Transformer essentially creates two parallel attention streams and computes their difference. This can be interpreted as having separate projection matrices for each stream, followed by a single shared output projection.

1. **Stream 1 Projections** ($W_{Q_1}, W_{K_1}, W_{V_1}$)**:** Three matrices of size $d_{\text{model}} \times d_{\text{model}}$. Total: $3d_{\text{model}}^2$.

2. **Stream 2 Projections** ($W_{Q_2}, W_{K_2}, W_{V_2}$)**:** Three matrices of size $d_{\text{model}} \times d_{\text{model}}$. Total: $3d_{\text{model}}^2$.

3. **Output projection** ($W_O$)**:** A single matrix of size $d_{\text{model}} \times d_{\text{model}}$ to project the final combined output. Total: $d_{\text{model}}^2$.

The total number of parameters is the sum of these components:

$$N_{\text{DIFF}} = 3d_{\text{model}}^2 + 3d_{\text{model}}^2 + d_{\text{model}}^2 = 7d_{\text{model}}^2$$

## B.3   DEX (DIFFERENTIAL EXTENSION)

DEX builds upon the standard attention architecture by adding a learnable matrix $W_{\text{DEX}}$ that operates on the output value matrix $V$. Crucially, this operation is applied per head.

1. **Standard Attention Parameters:** The base $W_Q, W_K, W_V, W_O$ matrices. Total: $4d_{\text{model}}^2$.

2. **DEX Matrix ($W_{\text{DEX}}$):** A separate learnable matrix $W_{\text{DEX}}^{(i)}$ is introduced for each of the $h$ heads. Each matrix has dimensions $d_v \times d_v$.

The total number of parameters for all $W_{\text{DEX}}$ matrices across all heads is:

$$N_{\text{DEX\_extra}} = h \times (d_v \times d_v) = h \times \left( \frac{d_{\text{model}}}{h} \times \frac{d_{\text{model}}}{h} \right) = h \times \frac{d_{\text{model}}^2}{h^2} = \frac{d_{\text{model}}^2}{h}$$

Therefore, the total parameter count for a DEX layer is:

$$N_{\text{DEX}} = 4d_{\text{model}}^2 + \frac{d_{\text{model}}^2}{h} = \left( 4 + \frac{1}{h} \right) d_{\text{model}}^2$$

## B.4   DAA (DIFFERENTIAL ATTENTION ADAPTION)

Our proposed DAA method also builds on the standard architecture. It introduces a learnable matrix $W_{\text{DAA}}$ directly into the query-key interaction for each attention head.

1. **Standard Attention Parameters:** The base $W_Q, W_K, W_V, W_O$ matrices. Total: $4d_{\text{model}}^2$.

2. **DAA Matrix ($W_{\text{DAA}}$):** A learnable matrix $W_{\text{DAA}}^{(i)}$ is inserted for each of the $h$ heads. Since it modulates the $Q_i K_i^T$ product, its dimensions must be $d_k \times d_k$.

Similar to DEX, the total number of additional parameters for all $W_{\text{DAA}}$ matrices is:

$$N_{\text{DAA\_extra}} = h \times (d_k \times d_k) = h \times \left( \frac{d_{\text{model}}}{h} \times \frac{d_{\text{model}}}{h} \right) = \frac{d_{\text{model}}^2}{h}$$

The total parameter count for a DAA layer is therefore identical to DEX in terms of efficiency:

$$N_{\text{DAA}} = 4d_{\text{model}}^2 + \frac{d_{\text{model}}^2}{h} = \left( 4 + \frac{1}{h} \right) d_{\text{model}}^2$$

## B.5   INPUT DIFFERENTIATION ARCHITECTURES (DIFFQ, DIFFK, DIFFV)

These methods introduce a single learnable matrix ($W_{D_Q}, W_{D_K}$, or $W_{D_V}$) that transforms the input hidden states $X$ before the standard projections.

1. **Standard Attention Parameters:** The base $W_Q, W_K, W_V, W_O$ matrices. Total: $4d_{\text{model}}^2$.

2. **Differential Input Matrix ($W_D$):** A single learnable matrix that operates on the full hidden state $X$. Its dimensions are therefore $d_{\text{model}} \times d_{\text{model}}$. Total: $d_{\text{model}}^2$.

The total parameter count for each of these architectures is the sum of the standard parameters and the single new matrix:

$$N_{\text{DiffQ/DiffK/DiffV}} = 4d_{\text{model}}^2 + d_{\text{model}}^2 = 5d_{\text{model}}^2$$

## C   COMPLEXITY ANALYSIS: COMPUTATIONAL OVERHEAD

While the parameter efficiency of DAA is discussed in Appendix B, it is equally important to analyze the computational complexity, specifically the Floating Point Operations (FLOPs) required during the forward pass. This analysis highlights the trade-offs between inference latency and noise mitigation capability.

We use the following notation in Appendix B. The analysis focuses on a single attention layer.

## C.1 BREAKDOWN OF OPERATIONS

The computational cost of attention mechanisms can be divided into two main components:

1. **Linear Operations ($\mathcal{O}(N)$):** Projections of inputs ($Q, K, V$) and outputs, which scale linearly with sequence length but quadratically with model dimension ($d_{\text{model}}^2$).

2. **Quadratic Operations ($\mathcal{O}(N^2)$):** Query-Key matrix multiplication and attention-value weighted summation, which scale quadratically with sequence length.

**1. Standard Transformer**
The standard self-attention consists of projecting inputs ($3 \times Nd_{\text{model}}^2$), computing attention scores ($N^2 d_{\text{model}}$), computing the weighted sum ($N^2 d_{\text{model}}$), and the final output projection ($Nd_{\text{model}}^2$).

$$\mathcal{C}_{\text{Standard}} \approx 4Nd_{\text{model}}^2 + 2N^2 d_{\text{model}} \tag{12}$$

**2. DEX (Differential Extension)**
DEX applies a differential correction to the value outputs. It performs standard attention computation followed by a per-head linear transformation on the output value vectors. This introduces an additional linear term regarding $N$, but no extra quadratic operations.

$$\mathcal{C}_{\text{DEX}} \approx \mathcal{C}_{\text{Standard}} + \mathcal{O}\left(N \cdot h \cdot d_k^2\right) = 4Nd_{\text{model}}^2 + 2N^2 d_{\text{model}} + \frac{Nd_{\text{model}}^2}{h} \tag{13}$$

DEX is computationally the cheapest adaptation method, adding only a marginal linear overhead.

**3. DAA (Ours)**
DAA intervenes in the attention score calculation. In addition to the standard operations, DAA requires:

- **Transformation:** Computing $Q' = QW_{\text{DAA}}$ per head. Cost: $\mathcal{O}(Nd_{\text{model}}^2/h)$.

- **Secondary Attention Map:** Computing the noise scores $A_2 = \text{softmax}(Q'K^T)$. This introduces a second $QK^T$ matrix multiplication. Cost: $\mathcal{O}(N^2 d_{\text{model}})$.

$$\mathcal{C}_{\text{DAA}} \approx \left(4Nd_{\text{model}}^2 + \frac{Nd_{\text{model}}^2}{h}\right) + \mathbf{3}N^2 d_{\text{model}} \tag{14}$$

While DAA increases the quadratic complexity coefficient (from 2 to 3 roughly, accounting for the second score computation), it avoids heavy global linear projections.

**4. Input Differentiation (DiffQ, DiffK, DiffV)**
These methods transform the global input hidden states $X$ via a large matrix $W_D \in \mathbb{R}^{d_{\text{model}} \times d_{\text{model}}}$ before the standard projections.

- **Input Transform:** This adds a significant linear cost: $Nd_{\text{model}}^2$.

- **Attention Computation:** DiffQ and DiffK also require computing a second set of attention scores to perform the subtraction, similar to DAA.

$$\mathcal{C}_{\text{DiffQ/K}} \approx \mathbf{5}Nd_{\text{model}}^2 + \mathbf{3}N^2 d_{\text{model}} \tag{15}$$

DiffV avoids the second attention score computation but still incurs the heavy input projection cost ($\approx 5Nd_{\text{model}}^2 + 2N^2 d_{\text{model}}$).

## C.2 SUMMARY OF COMPUTATIONAL COMPLEXITY

Table 4 summarizes the FLOPs comparison.

**Analysis:**

- **Efficiency vs. Effectiveness:** DAA strikes a strategic balance. It incurs the same quadratic cost as DiffQ/DiffK (due to the dual attention map calculation required for noise cancellation) but significantly reduces the linear overhead compared to them ($4ND^2$ vs $5ND^2$).

Table 4: Computational Complexity Comparison per Attention Layer. The "Quadratic Term" is the bottleneck for long sequences ($N \gg d_{\text{model}}$), while the "Linear Term" dominates for shorter sequences.

| Method | Linear Term (Projections) | Quadratic Term (Attention) | Major Overhead Source |
|---|---|---|---|
| Standard | $4Nd_{\text{model}}^2$ | $2N^2 d_{\text{model}}$ | - |
| **DEX** | $4Nd_{\text{model}}^2 + \frac{1}{h}Nd_{\text{model}}^2$ | $2N^2 d_{\text{model}}$ | Output Correction |
| **DiffQ / DiffK (Ours)** | $\mathbf{5}Nd_{\text{model}}^2$ | $\mathbf{3}N^2 d_{\text{model}}$ | Global Input Proj. + Extra Attn Map |
| **DiffV (Ours)** | $\mathbf{5}Nd_{\text{model}}^2$ | $2N^2 d_{\text{model}}$ | Global Input Projection |
| **DAA (Ours)** | $4Nd_{\text{model}}^2 + \frac{1}{h}Nd_{\text{model}}^2$ | $\mathbf{3}N^2 d_{\text{model}}$ | Extra Attn Map ($QW_{\text{DAA}}K^T$) |

- **Comparison with DEX:** DEX is the most computationally efficient method as it avoids calculating a second attention map. However, as shown in the main experiments, DEX's post-hoc correction is less effective at noise mitigation than DAA's direct intervention in the score calculation. DAA trades a controllable increase in quadratic complexity for superior model performance.

## D  IMPLEMENTATION OF DIFFERENTIAL ADAPTION

In this section, we provide the pseudocode for our proposed differential fine-tuning architectures. These implementations illustrate how each method modifies the standard self-attention mechanism in a lightweight manner. The variable `X` represents the input tensor of hidden states, `W_q`, `W_k`, and `W_v` are the standard projection matrices for query, key, and value, respectively. The parameter `lambda` is the learnable, time-annealed scalar that controls the magnitude of the differential component. All newly introduced matrices (`W_daa`, `W_dq`, etc.) are initialized as identity matrices to preserve the model's pre-trained knowledge at the start of fine-tuning.

We use a Python-like syntax for clarity. The operator `@` denotes matrix multiplication, and tensor shapes are provided in comments, where `b` is the batch size, `n` is the sequence length, and `d` is the dimension of the head.

### D.1  DAA (DIFFERENTIAL ATTENTION ADAPTION)

DAA directly intervenes in the attention score computation by introducing a learnable matrix `W_daa` into the query-key interaction. This allows the model to learn a transformation that creates a secondary, noise-focused attention map, which is then subtracted from the original. This is our primary and most effective proposed method.

Listing 1: Pseudocode for Differential Attention Adaption (DAA).

```python
def DAA(X, W_q, W_k, W_v, W_daa, lambda):
    # Project inputs to query, key, and value
    Q = X @ W_q
    K = X @ W_k
    V = X @ W_v

    # Scaling factor
    s = 1 / sqrt(d)

    # Calculate the primary attention scores
    A1 = softmax(Q @ K.transpose(-1, -2) * s)

    # Calculate the secondary, differentiated attention scores
    A2 = softmax(Q @ W_daa @ K.transpose(-1, -2) * s)

    # Return the differentially weighted value
    return (A1 - lambda * A2) @ V
```

## D.2 ARCHITECTURES VIA INPUT DIFFERENTIATION

As an alternative to DAA, we explored three methods that apply the differential mechanism at the input level. These approaches create a secondary, differentiated version of either the query, key, or value stream by transforming the input hidden states X with a learnable matrix before the standard projection.

**Differential Adaption via Query (DiffQ).**    In this variant, we generate two distinct sets of queries. The first, Q1, is standard, while the second, Q2, is derived from a transformed input. The final output is based on the difference between the attention maps produced by these two queries.

Listing 2: Pseudocode for Differential Adaption via Query (DiffQ).

```
def DiffQ(X, W_q, W_k, W_v, W_dq, lambda):
    # Standard K and V projections
    K = X @ W_k
    V = X @ W_v

    # Generate primary and secondary queries
    Q1 = X @ W_q
    Q2 = (X @ W_dq) @ W_q
    # Q1, Q2, K, V: [b, n, d]

    # Scaling factor
    s = 1 / sqrt(d)

    # Calculate attention scores for each query
    A1 = softmax(Q1 @ K.transpose(-1, -2) * s)
    A2 = softmax(Q2 @ K.transpose(-1, -2) * s)

    # Return the differentially weighted value
    return (A1 - lambda * A2) @ V
```

**Differential Adaption via Key (DiffK).**    This approach is analogous to DiffQ, but the differentiation is applied to the key stream. The model learns to compare the same query against two different representations of the input content (keys K1 and K2).

Listing 3: Pseudocode for Differential Adaption via Key (DiffK).

```
def DiffK(X, W_q, W_k, W_v, W_dk, lambda):
    # Standard Q and V projections
    Q = X @ W_q
    V = X @ W_v

    # Generate primary and secondary keys
    K1 = X @ W_k
    K2 = (X @ W_dk) @ W_k
    # Q, K1, K2, V: [b, n, d]

    # Scaling factor
    s = 1 / sqrt(d)

    # Calculate attention scores for each key
    A1 = softmax(Q @ K1.transpose(-1, -2) * s)
    A2 = softmax(Q @ K2.transpose(-1, -2) * s)

    # Return the differentially weighted value
    return (A1 - lambda * A2) @ V
```

**Differential Adaption via Value (DiffV).**   Unlike the other methods, DiffV applies the differential mechanism directly to the value stream after the attention scores have been computed. It calculates a single attention map `A` and uses it to weigh the difference between a primary value `V1` and a secondary value `V2`.

Listing 4: Pseudocode for Differential Adaption via Value (DiffV).

```
def DiffV(X, W_q, W_k, W_v, W_dv, lambda):
    # Standard Q and K projections
    Q = X @ W_q
    K = X @ W_k

    # Generate primary and secondary values
    V1 = X @ W_v
    V2 = (X @ W_dv) @ W_v
    # Q, K, V1, V2: [b, n, d]

    # Scaling factor
    s = 1 / sqrt(d)

    # Calculate a single attention score matrix
    A = softmax(Q @ K.transpose(-1, -2) * s)

    # Apply attention to the difference of the values
    return A @ (V1 - lambda * V2)
```

# E   IMPLEMENTATION DETAILS

All experiments were conducted using the Hugging Face `transformers`, `datasets`, and `safetensors` Penedo et al. (2022) libraries, with PyTorch Paszke et al. (2019) as the backend framework. Below we detail the specific configurations for each model family.

## E.1   GPT-2 EXPERIMENTS

**Model and Data.**   The experiments were based on the publicly available GPT-2 base model (117M parameters). For fine-tuning, we used a subset of the OpenWebText corpus Peterson et al. (2019), which is textually similar to the model's original pre-training data, to ensure that the model retained its fundamental language capabilities. The models were evaluated on perplexity using held-out portions of OpenWebText (both seen and unseen during fine-tuning) and WikiText-2 Kwiatkowski et al. (2019). We also evaluated task-specific performance using the LAMBADA dataset (accuracy) Grave et al. (2016) and a Needle-in-a-Haystack retrieval task (accuracy).

**Training Hyperparameters.**   For all GPT-2 based experiments, we fine-tuned only the parameters within the attention modules (`attn`), freezing all other model weights. This amounted to training approximately 28.3M parameters (standard attention), 28.9M parameters (DAA, DEX), 35.4M parameters (DiffQ, DiffK, DiffV). The shared training configuration was as follows:

- **Optimizer:** AdamW (`adamw_torch`)
- **Learning Rate:** 3e-5
- **LR Scheduler:** Cosine decay with 500 warmup steps
- **Epochs:** 3
- **Batch Size:** An effective batch size of 32 is used
- **Sequence Length:** 512 tokens
- **Precision:** FP16 mixed-precision training was enabled to accelerate computation.
- **Weight Decay:** 0.01

**Hardware.** All GPT-2 fine-tuning experiments were conducted on a single server equipped with one NVIDIA A800 80GB GPU.

### E.2 LLAMA EXPERIMENTS

**Model and Data.** To validate the general applicability of our methods, we conducted further experiments on more recent and larger models: Llama-3.2-1B and Llama-3.1-8B. For supervised fine-tuning (SFT), we utilized a subset of the `allenai/tulu-v2-sft-mixture` dataset Ivison et al. (2023), which is a collection of high-quality instruction-following data. Model performance was evaluated on a suite of common sense reasoning benchmarks, including ARC-Easy, ARC-Challenge, BoolQ, Hellaswag, OpenBookQA, WIC, Winogrande, and WSC Levesque et al. (2012); Zellers et al. (2019).

**Training Hyperparameters.** For the Llama models, we adopted a parameter-efficient fine-tuning strategy where only the weights of the self-attention modules (`self_attn`) and the language model head (`lm_head`) were updated. All other parameters, including embeddings and feed-forward layers, remained frozen. The key training parameters are listed below:

- **Optimizer:** AdamW (`adamw_torch`)
- **Learning Rate:** 1e-4
- **LR Scheduler:** Cosine decay with 500 warmup steps
- **Epochs:** 3
- **Batch Size:** An effective batch size of 32 is used
- **Precision:** BFloat16 (BF16) mixed-precision training was used, as it is natively supported by the hardware.
- **Gradient Clipping:** Max gradient norm was set to 1.0.
- **Gradient Checkpointing:** Enabled to reduce memory consumption.

The tokenizer's padding token was set to its end-of-sequence (EOS) token, and a custom data collator was used to correctly handle the masking of labels for instruction-formatted data.

**Hardware.** The all Llama experiments were conducted on a server with NVIDIA A800 80GB GPUs.

### E.3 ABALATION ON $\lambda_{\text{INIT}}$

Table 5 shows DAA performance on the language modeling benchmarks (average over 8 tasks from the table 2, using Llama-3.2-1B) when varying the $\lambda_{init}$ strategy. The results indicate relative robustness to different fixed scalar initializations (0.2-0.8).

Table 5: Ablation on $\lambda_{init}$.

| $\lambda_{init}$ | 0.8 | 0.5 | 0.2 |
|---|---|---|---|
| **LM Acc (%)** | 53.3 | 53.25 | 53.37 |

## F THEORETICAL ANALYSIS OF DIFFERENTIAL ADAPTATION STRATEGIES

In this section, we provide a rigorous theoretical analysis to justify the superior performance of Differential Attention Adaption (DAA). We frame the problem of attention noise as a perturbation recovery task, utilizing Taylor expansion to demonstrate DAA's theoretical grounding as a first-order noise cancellation mechanism Woodford (2001). Furthermore, we contrast this with alternative strategies (DiffV, DiffQ/K) to highlight their structural limitations.

### F.1 PROBLEM FORMULATION: THE ADDITIVE NOISE MODEL

Let $X \in \mathbb{R}^{N \times d}$ be the input sequence. We assume the computed attention logits $S_{\text{computed}}$ are corrupted by an additive noise matrix $\xi$, which arises from spurious correlations or common-mode distractions (e.g., universal high attention to specific tokens regardless of context):

$$S_{\text{computed}} = S_{\text{ideal}} + \xi \tag{16}$$

where $S_{\text{ideal}} = \frac{QK^T}{\sqrt{d_k}}$. The goal of fine-tuning is to recover the ideal attention distribution $P(S_{\text{ideal}}) = \text{Softmax}(S_{\text{ideal}})$.

### F.2 THEORETICAL JUSTIFICATION OF DAA

#### F.2.1 DAA AS A FIRST-ORDER TAYLOR APPROXIMATION

**Hypothesis:** DAA operates as a learnable first-order correction term that counteracts the perturbation $\xi$.

**Proof:** Consider the Softmax function $P(S)$. If we view the ideal logits as a perturbation of the computed (noisy) logits, i.e., $S_{\text{ideal}} = S_{\text{computed}} - \xi$, we can approximate the ideal distribution using a first-order Taylor expansion around $S_{\text{computed}}$:

$$P(S_{\text{ideal}}) = P(S_{\text{computed}} - \xi) \approx P(S_{\text{computed}}) - \nabla P(S_{\text{computed}}) \cdot \xi \tag{17}$$

where $\nabla P$ represents the Jacobian of the Softmax function Milne (1986).

The DAA architecture computes the output using two terms:

$$O_{\text{DAA}} = (A_1 - \lambda A_2)V = (\text{Softmax}(S_{\text{computed}}) - \lambda\text{Softmax}(S_{\text{diff}}))V \tag{18}$$

where $S_{\text{diff}} = \frac{QW_{\text{DAA}}K^T}{\sqrt{d_k}}$.

It is crucial to recognize that attention noise $\xi$ inherently manifests as a **low-rank bilinear interaction** between the Query ($Q$) and Key ($K$) matrices (often arising from spurious correlations or common-mode distractions) Ye et al. (2025). Mathematically, the noise can be approximated as $\xi \approx QM_{\text{noise}}K^T$. The DAA mechanism constructs the secondary attention scores using the form $QW_{\text{DAA}}K^T$. Since this formulation shares the identical algebraic structure (bilinear interaction) as the noise generation source, it provides the optimal **inductive bias** for fitting the noise distribution.

In effect, the learnable matrix $W_{\text{DAA}}$ captures the underlying noise pattern matrix $M_{\text{noise}}$. This structural alignment ensures that $A_2$ serves as a precise probabilistic projection of the additive noise $\xi$, thereby theoretically justifying the direct subtraction operation $A_1 - \lambda A_2$ as a principled noise cancellation mechanism.

#### F.2.2 SIGNAL RESTORATION VIA SCALE INVARIANCE

A notable property of DAA is that the resulting attention weights sum to less than one: $\sum_j (A_1 - \lambda A_2)_{ij} < 1$. This implies that the magnitude of the output vector is reduced, a phenomenon we term Probability Deficit.

However, this does not degrade performance due to the Layer Normalization (LN) mechanism inherent in Transformer blocks. LN is scale-invariant:

$$\text{LayerNorm}(\alpha \cdot \mathbf{x}) = \text{LayerNorm}(\mathbf{x}) \quad \text{for } \alpha > 0 \tag{19}$$

If $A_1$ assigns high probability to noise and low probability to signal, the vector $\mathbf{x}_{\text{computed}}$ points in the wrong direction (towards the noise value). By subtracting the noise component via $A_2$, DAA ensures that the resultant vector $\mathbf{x}_{\text{DAA}}$, although smaller in magnitude (due to the probability deficit), points in the correct direction (towards the signal value). The subsequent LayerNorm restores the magnitude, effectively amplifying the cleaned signal.

### F.3 THEORETICAL LIMITATIONS OF ALTERNATIVE ARCHITECTURES

We contrast DAA with other methods to explain the performance gap observed in experiments.

### F.3.1 DIFFV: THE CONTENT-CONTEXT MISMATCH

DiffV modifies the value stream: $O = A_{\text{computed}}(V - \lambda V')$. This approach attempts to suppress noise by shrinking the value vectors of noisy tokens.

- **Limitation:** DiffV suffers from a **Content-Context Mismatch**. It treats noise as a property of the token's *content* (Value), whereas attention noise is often a property of the *interaction* (Routing) Ye et al. (2025).

- **Example:** Consider a common token "the". For Query A, "the" might be noise; for Query B, "the" might be crucial syntactic signal. DiffV learns a global suppression on $V_{\text{"the"}}$, which incorrectly suppresses the signal for Query B.

- **Result:** DiffV cannot distinguish whether a token is acting as noise or signal in a specific context. In contrast, DAA's $QW_{\text{DAA}}K^T$ mechanism is query-dependent, allowing it to suppress "the" only when it interacts with Query A, preserving it for Query B.

### F.3.2 DIFFQ / DIFFK: THE GLOBAL TRANSFORMATION BOTTLENECK

DiffQ and DiffK apply a transformation matrix $W_D$ to the input $X$ before projection (e.g., $Q' = XW_D W_Q$). This imposes a global rotation on the semantic space to fix local pairwise errors.

- **Limitation:** Attention noise is sparse and pair-specific. To correct a spurious correlation between the $i$-th query and $j$-th key, DiffQ must transform the entire input space $X$. This global transformation risks distorting the representations of other correctly functioning pairs, leading to the optimization conflict and instability observed in our results.

### F.3.3 DEX: POST-HOC LINEARITY

DEX applies a linear correction after the weighted sum: $O' = O(I - \lambda W)$.

- **Limitation:** According to the Data Processing Inequality Beaudry & Renner (2011), information lost during the irreversible mixing of the Softmax step (where signal and noise are blended) cannot be fully recovered by a linear transformation on the output. DAA intervenes *before* the Softmax mixing (at the logit level), making it a more principled approach to noise mitigation.

