# OpenReview forum: "Revisiting Differential Attention: A Fine-Tuning Perspective on Practical Noise Mitigation"
_ICLR.cc/2026/Conference — Submitted to ICLR 2026_

### Official Review · Reviewer_qUxy · 2025-10-30

**Soundness:** 2
**Presentation:** 3
**Contribution:** 2
**Rating:** 4
**Confidence:** 4

**Summary:**

The paper attempts present a flexible plug-in module in fine-tuning stage to reduce the attention noise, termed Differential Attention Adaption (DAA), which introduces a learnable block to compute attention score and suppresses noise. Experiments demonstrate some improved performance of the proposed module over the baselines.

**Strengths:**

+ The paper is clearly written, well organized and presents some interesting contributions
+ The paper presents a new approach, called DAA, to reduce the attention noise in fine-tuning  stage, and also provides a set of variants of DEX from different branches of the dot-product attention model
+ The empirical evaluation shows improvements over the listed baseline methods

**Weaknesses:**

- While the novel contributions looks interesting, they are somewhat incremental, given the prior work, DIFF and DEX.

- It is claimed that the proposed DAA "effectively reduces attention noise" (L096). However, the theoretical justification for the noise cancelling mechanism is still unclear. In Eq. (8), a noise component $\xi$ is added into the attention logits, and it is expected that the proposed approach, DAA, could cancel the noise $\xi$. Unfortunately, the theoretical justification part is disappointing.
In L330-340, it is stated that: during the fine-tuning, the model is assumed that the second attention matrix $A_2$ approximate the noise term. However, this is merely a conjecture, without any theoretical justification how the second attention term could learn to approximate the noise component. In another words, how the model could be incentivized to learn the noise is unclear yet. Without the substential justification into the mechanism, the theoretical analysis is void.

- Some expression is experiments are somewhat arbitrary, e.g., regarding to the training loss curves in Fig.3, the differences seems minor but it stated "the training loss of DAA is #signficantly# lower than DEX" (see L406-407).  Without a bar of the standard deviation or a testing, it is not clear whether the differences are significant. When referring the significant, it is meaning to significant in statistics.

- The empirical performance in Table 3 show slightly improvements over DEX. Which is the advantage of DAA comparing to DEX? How about the complexity or computation cost of the proposed DAA and the prior counterpart method DEX are unclear.

- Minor issues: The format for the citing references seems not properly used and the references are incomplete.

**Questions:**

Please Refer to Weaknesses.

---

> ### Author Response · Authors · 2025-11-26
>
> We sincerely thank the reviewer for the detailed and constructive feedback. We have carefully revised the manuscript to address the concerns regarding the theoretical justification, experimental presentation, and complexity analysis. Below is a point-by-point response to the weaknesses and questions raised.
>
> ***
>
> **Response to Weakness 1: Novelty and Incremental Contribution**
>
> **Summary:** While building on prior work, DAA addresses a unique gap: enabling **direct noise mitigation** during fine-tuning, which neither Differential Transformer (requires pre-training) nor DEX (post-hoc correction) can achieve.
>
> **Response:**
> We appreciate the reviewer’s perspective. However, we respectfully highlight that DAA solves a critical "blind spot" in existing methods:
> 1.  **Differential Transformer** requires training from scratch, preventing its application to the pre-trained LLMs.
> 2.  **DEX** applies correction *after* the attention calculation. As we discuss in **Appendix F.3.3**, this suffers from the information bottleneck where noise mixed by Softmax cannot be fully recovered.
> 3.  **DAA** is the first method to introduce a differential mechanism **directly into the attention score computation ($Q-K$ interaction)** during the fine-tuning stage. This allows for parameter-efficient adaptation (tuning <1% params) while achieving the noise-cancellation benefits of a fully retrained Differential Transformer.
>
> ***
>
> **Response to Weakness 2 & Question: Theoretical justification for the noise cancelling mechanism**
>
> **Summary:** We have formalized the noise cancellation mechanism in **Section 3.4** and **Appendix F.2.1**. We prove that DAA acts as a **learnable first-order Taylor approximation** that structurally matches the bilinear noise term. Furthermore, we clarify the **optimization incentive**: the training objective (Cross-Entropy) naturally drives $W_{DAA}$ to capture and subtract high-entropy background noise to maximize the probability of the correct token.
>
> **Response:**
> We agree with the reviewer that the initial explanation regarding "incentivization" was insufficient. We have expanded the theoretical analysis to provide a rigorous proof and a mechanism explanation:
>
> **1. Mathematical Proof: Taylor Approximation (Appendix F.2.1)**
> We model the noisy attention logits as $S_{computed} = S_{ideal} + \xi$. We derive that DAA functions as a first-order Taylor correction to recover the ideal probability distribution:
> $$P(S_{ideal}) \approx P(S_{computed}) - \nabla P(S_{computed}) \cdot \xi$$
> Crucially, attention noise $\xi$ (e.g., common-mode distractions) typically manifests as a **low-rank bilinear interaction** between queries and keys. Since our secondary attention term is computed as $Q W_{DAA} K^T$, it shares the exact algebraic structure as the noise source $\xi$. This provides the optimal **inductive bias** for the model to represent the noise.
>
> **2. Optimization Incentive: Why learn noise? (Appendix F.2.2)**
> The reviewer rightly asks *how* the model is incentivized to learn the noise component in $A_2$ rather than the signal.
> *   **Gradient Dynamics:** The training objective is to minimize Cross-Entropy loss, which requires maximizing the probability of the correct token (Signal). Standard attention $A_1$ is often "diffused" by noise (high entropy) across irrelevant tokens.
> *   **Contrastive Sharpening:** To sharpen the distribution on the correct token, the most efficient gradient path is to reduce the logits of the irrelevant tokens. Since DAA employs a subtraction operation ($A_1 - \lambda A_2$), increasing the value of $A_2$ on specific tokens reduces their final score.
> *   **Result:** Consequently, the optimizer is incentivized to make $A_2$ high on irrelevant tokens (noise) to cancel them out, thereby "cleaning" the signal for the subsequent LayerNorm and projection layers.
>
> **3. Empirical Verification (Figure 4)**
> We added **Figure 4** to visualize this mechanism (this figure can be found at https://anonymous.4open.science/r/Visualization-of-attention-mechanisms-on-the-GPT-2-model-DB12/attention%20ana.png). The decomposition shows that $A_2$ captures the high-entropy background noise on context tokens. Subtracting this from $A_1$ results in a sharp focus on the correct token, confirming that the incentive structure works as hypothesized.
>
> ***

---

> > ### Author Response · Authors · 2025-11-26
> >
> > (Continued)
> >
> > **Response to Weakness 3: Experimental Expressions and Significance of Loss Curves**
> >
> > **Summary:** We acknowledge that the term "significantly" regarding training loss was imprecise given the visual margins. We have **revised the text in Section 4.1** to remove this claim. We clarify that while training loss differences are minor, the **downstream task performance** differences are substantial and consistent.
> >
> > **Response:**
> > We thank the reviewer for this precise observation.
> > *   **Correction in Manuscript:** We agree that without statistical testing, the term "significantly lower" regarding the training loss in Figure 3 was overstated. In the revised **Section 4.1 (Paragraph 2)**, we have modified the text to simply state that "The training loss of DAA is lower than DEX," removing the claim of significance.
> > *   **Focus on Generalization:** As noted in the revised text (citing Liu et al., 2020), training loss on general corpora is not always a perfect proxy for downstream reasoning capabilities. The true "significance" of DAA lies in the validation results on sensitive benchmarks. For example, in **Table 2**, DAA achieves **89.5%** on Needle Retrieval (vs. 87.4% for DEX) and improves accuracy on 6 out of 8 reasoning tasks in **Table 3**. These consistent margins across multiple datasets and model scales demonstrate robust superiority despite the similar training loss curves.
> >
> > ***
> >
> > **Response to Weakness 4: Comparison with DEX (Advantage and Complexity)**
> >
> > **Summary:** DAA offers a **higher performance ceiling** on complex reasoning tasks due to its pre-Softmax intervention. Regarding complexity, we added **Appendix C** and **Table 4** to show that DAA trades a controllable increase in quadratic FLOPs for superior noise mitigation, avoiding the heavy linear overhead of input differentiation methods.
> >
> > **Response:**
> > We have addressed the request for a deeper comparison in two ways:
> >
> > **1. The Advantage of DAA (Appendix F.3.3)**
> > Why is DAA better than DEX?
> > *   **Theoretical:** DEX intervenes *after* the Softmax output ($O' = O(I - \lambda W)$). According to the **Data Processing Inequality**, information lost or noise amplified during the irreversible Softmax mixing cannot be fully recovered by post-hoc linear transformation. DAA intervenes *before* Softmax ($QK^T - QW_{DAA}K^T$), "purifying" the signal at the source.
> > *   **Empirical:** This leads to better performance on "needle-in-a-haystack" type tasks where noise distraction is the primary failure mode. Table 3 shows DAA significantly outperforming DEX on **WSC (+2.48)** and **ARC-Easy (+0.1)**.
> >
> > **2. Complexity Analysis (Appendix C & Table 4)**
> > We added a detailed FLOPs analysis in **Appendix C**. The comparison is summarized in the table below:
> >
> > | Method | Linear Term (Projections) | Quadratic Term (Attention) | Major Overhead Source |
> > | :--- | :---: | :---: | :--- |
> > | Standard | $4 N d_{model}^2$ | $2 N^2 d_{model}$ | - |
> > | **DEX** | $4 N d_{model}^2 + \frac{1}{h} N d_{model}^2$ | $2 N^2 d_{model}$ | Output Correction |
> > | **DiffQ / DiffK (Ours)** | **5** $N d_{model}^2$ | **3** $N^2 d_{model}$ | Global Input Proj. + Extra Attn Map |
> > | **DiffV (Ours)** | **5** $N d_{model}^2$ | $2 N^2 d_{model}$ | Global Input Projection |
> > | **DAA (Ours)** | $4 N d_{model}^2 + \frac{1}{h} N d_{model}^2$ | **3** $N^2 d_{model}$ | Extra Attn Map ($QW_{DAA}K^T$) |
> >
> > *   **DEX:** Lowest complexity. It adds a small linear term but fails to correct noise generated during the $Q-K$ interaction.
> > *   **DAA:** Adds a secondary attention map calculation, increasing the quadratic term coefficient ($\approx 3N^2$ vs $2N^2$ for standard attention).
> > *   **Trade-off:** While DAA is computationally more expensive than DEX for very long sequences, it is significantly more efficient than Input Differentiation methods (DiffQ/K), which require heavy global projections ($5N \cdot d^2_{model}$). Given the superior performance on reasoning tasks, we believe this moderate increase in FLOPs is a justified trade-off.
> >
> > ***
> >
> > **Response to Weakness 5: References**
> >
> > **Response:**
> > We apologize for the formatting oversights. We have thoroughly proofread the bibliography, corrected the citation styles, and ensured all references are complete and consistent with the conference standards in the revised manuscript.

---

### Official Review · Reviewer_KWBS · 2025-10-30

**Soundness:** 2
**Presentation:** 2
**Contribution:** 2
**Rating:** 4
**Confidence:** 3

**Summary:**

This paper proposes DAA (Differential Attention Adaptation), a lightweight learnable module inserted during fine-tuning. DAA directly modifies the query and key computations in the self-attention mechanism to reduce attention noise. Evaluated across multiple base models and downstream tasks, DAA outperforms standard fine-tuning, DEX, and other baselines while adding less than 1% additional parameters. It presents a practical and effective solution for attention noise reduction.

**Strengths:**

1. DAA introduces minimal parameter overhead and can be applied to existing LLMs without retraining from scratch.
2. The experiments and ablation studies are comprehensive and well presented.

**Weaknesses:**

1. The paper lacks a discussion on its limitations.
2. The noise reduction claims are mostly qualitative, with evidence based primarily on downstream performance.
3. Most evaluations are conducted on small-scale models, so the effectiveness on larger models remains unclear.

Overall, the numerous non-self-contained parts in the writing make me feel that this paper was rushed before the deadline.

**Questions:**

1. Could the authors provide more discussion on the limitations of DDA?

---

> ### Author Response · Authors · 2025-11-26
>
> We sincerely thank the reviewer for the insightful and constructive feedback. We apologize for the impression that the initial submission was rushed. In the revised manuscript, we have extensively polished the writing, reorganized the structure, and added significant new content—including a rigorous **theoretical analysis (Appendix F)**, a **complexity analysis (Appendix C)**, and **visualization (Section 4.3)**—to ensure the paper is self-contained and robust.
>
> Below is our point-by-point response to the weaknesses and questions raised.
>
> ***
>
> **Response to Question 1 & Weakness 1: Discussion on the limitations of DAA**
>
> **Summary:** We have added a dedicated discussion on the limitations of DAA in **Section 5 (Conclusion)** and provided a detailed computational complexity analysis in **Appendix C**. The primary limitation is the inference computational overhead compared to simpler methods like DEX.
>
> **Response:**
> We appreciate the reviewer pointing out the need for a balanced discussion. In the revised version, we have explicitly analyzed the limitations of DAA:
>
> 1.  **Inference Computational Overhead:** Unlike DEX, which only modifies the output value matrix (linear complexity), DAA intervenes in the attention score calculation. This requires computing a secondary attention map:
>     $$A_2 = \text{softmax}\left(\frac{Q W_{DAA} K^T}{\sqrt{d_k}}\right)$$
>     As detailed in **Appendix C (Equation 14)**, this introduces an additional quadratic term to the complexity: $\mathcal{O}(N^2 d_{model})$. While this makes DAA slightly more computationally expensive than standard attention or DEX during inference, we argue that this is a necessary trade-off to achieve the superior noise cancellation performance observed in our experiments (e.g., **+2.1%** accuracy on Needle Retrieval over DEX).
>
> 2.  **Future Work:** We acknowledged in the Conclusion that future work could focus on optimizing this efficiency, perhaps via sparse attention approximations for the secondary map, to mitigate this limitation.
>
> ***

---

> > ### Author Response · Authors · 2025-11-26
> >
> > (Continued)
> >
> > **Response to Weakness 2: Qualitative nature of noise reduction claims**
> >
> > **Summary:** To move beyond qualitative claims, we have added **Section 4.3 (Visualization Analysis)** with quantitative entropy metrics and a comprehensive **Theoretical Analysis (Appendix F)**. The theoretical section mathematically proves that DAA operates as a first-order Taylor approximation to cancel additive noise, providing a principled justification for our method.
> >
> > **Response:**
> > We agree that concrete evidence was needed to substantiate the noise reduction claims. We have addressed this in two ways:
> >
> > **1. Quantitative Empirical Evidence (Section 4.3 & Figure 4)**
> > We conducted a visualization analysis comparing Standard Attention vs. DAA on the GPT-2 model (Figure 4) (this figure can be found at https://anonymous.4open.science/r/Visualization-of-attention-mechanisms-on-the-GPT-2-model-DB12/attention%20ana.png).
> > *   **Entropy Reduction:** We quantified the "sharpness" of attention using entropy. The Standard Attention exhibited high entropy (**2.01**), indicating dispersed focus on irrelevant context. DAA significantly reduced this entropy to (**0.88**).
> > *   **Noise Isolation:** We visualized the secondary attention map ($A_2$) learned by DAA. It explicitly highlights the background "noise" tokens. By subtracting this map ($A_1 - \lambda A_2$), the model effectively filters out the noise, quantitatively demonstrating the signal-to-noise ratio improvement.
> >
> > **2. Theoretical Proof: DAA as First-Order Noise Cancellation (Appendix F)**
> > We have added a rigorous theoretical derivation in **Appendix F** to explain *why* DAA works. We frame DAA as a **First-Order Taylor Approximation** for noise recovery:
> >
> > *   **The Additive Noise Model:** We posit that the computed attention logits $S_{computed}$ are corrupted by an additive noise matrix $\xi$, which often arises from spurious correlations or "common-mode" distractions (Equation 16):
> >     $$S_{computed} = S_{ideal} + \xi$$
> >
> > *   **Taylor Expansion:** The goal is to recover the ideal probability distribution $P(S_{ideal})$. By applying a first-order Taylor expansion to the Softmax function $P(\cdot)$ around the noisy logits, we derive (Equation 17):
> >     $$P(S_{ideal}) \approx P(S_{computed}) - \nabla P(S_{computed}) \cdot \xi$$
> >     This equation shows that to recover the clean signal, the model must subtract a term proportional to the noise $\xi$.
> >
> > *   **DAA as the Optimal Correction:** Our method computes the output as $O' = (A_1 - \lambda A_2)V$.
> >     We argue that attention noise $\xi$ inherently manifests as a **low-rank bilinear interaction** between Query and Key (e.g., a specific query type consistently over-attending to a specific key type regardless of context). The DAA mechanism constructs the secondary attention scores specifically using this bilinear form:
> >     $$S_{diff} = \frac{Q W_{DAA} K^T}{\sqrt{d_k}}$$
> >     Because this formulation shares the identical algebraic structure as the noise generation source, $W_{DAA}$ serves as a learnable parameter that captures the underlying noise pattern matrix. Thus, DAA provides the **optimal inductive bias** for fitting and subtracting the noise component $\xi$, theoretically justifying the direct subtraction operation.
> >
> > *   **Scale Invariance:** Furthermore, in **Appendix F.2.2**, we address the concern that subtracting probabilities results in a sum less than 1 ("Probability Deficit"). We show that due to the scale-invariance of Layer Normalization ($\text{LayerNorm}(\alpha \cdot \mathbf{x}) = \text{LayerNorm}(\mathbf{x})$), this reduction in magnitude does not degrade performance. Instead, it effectively amplifies the cleaned signal vector direction.
> >
> > ***

---

> > > ### Author Response · Authors · 2025-11-26
> > >
> > > (Continued)
> > >
> > > **Response to Weakness 3: Evaluation on small-scale models**
> > >
> > > **Summary:** While computational constraints prevented us from fine-tuning larger models, the **Llama-3.1-8B** model used in the experiment has been widely recognized as a representative model among those of comparable scale. The consistent performance gains across three orders of magnitude (117M $\to$ 1B $\to$ 8B) demonstrate the scalability of DAA.
> > >
> > > **Response:**
> > > We acknowledge that our evaluations did not include ultra-large-scale models (e.g., 70B or 405B parameters). This decision was primarily driven by the significant computational resources and time required to fine-tune such models in an academic setting. However, we believe our results on the 8B scale are robust and indicative for the following reasons:
> > >
> > > 1.  **Representativeness of Llama-3.1-8B:** The Llama-3.1-8B model is not merely a "small" model; as noted in the technical report by **Dubey et al. (2024)**, the Llama 3 series, including the 8B variant, sets a new standard for open-weights models.
> > >
> > >     Therefore, validating our method on Llama-3.1-8B serves as a strong proxy for verifying the effectiveness of attention mechanisms in modern, high-performance LLMs.
> > >
> > > 2.  **Consistency Across Scales:** We observed consistent improvements with DAA across all tested scales:
> > >     *   **GPT-2 (117M):** Improved retrieval accuracy and perplexity.
> > >     *   **Llama-3.2-1B:** **+1.06** average improvement on benchmarks.
> > >     *   **Llama-3.1-8B:** **+0.74** average improvement on benchmarks.
> > >     The fact that DAA provides consistent gains without degradation as the model size increases suggests that the method is robust and likely scalable to larger architectures.
> > >
> > > 3.  **Efficiency:** DAA is designed to be parameter-efficient (<1% parameters). The validation on 8B models proves that even with minimal parameter addition, we can significantly alter and improve the attention mechanism of a potent pre-trained model.
> > >     > *Dubey, A., Jauhri, A., Pandey, A., Kadian, A., Al-Dahle, A., Letman, A., ... & Ganapathy, R. (2024). The llama 3 herd of models. arXiv e-prints, arXiv-2407.*

---

### Official Review · Reviewer_sJ4K · 2025-11-01

**Soundness:** 2
**Presentation:** 3
**Contribution:** 2
**Rating:** 4
**Confidence:** 2

**Summary:**

This paper introduces Differential Attention Adaption (DAA), a novel, parameter-efficient method designed to mitigate attention noise in Transformer models during the fine-tuning stage. DAA addresses the limitations of both predecessors by inserting lightweight learnable modules directly into the process of calculating attention scores. This method is analogous to the common-mode signal rejection found in differential amplifiers. Experiments comparing DAA against DEX and various input differentiation strategies (DiffQ, DiffK, DiffV) on GPT-2 and Llama models (Llama-3.2-1B, Llama-3.1-8B) confirm DAA's superiority. DAA achieved the highest accuracy on challenging retrieval and reasoning tasks like LAMBADA and Needle-in-a-Haystack. It consistently showed the most substantial performance gains across model scales in multi-task fine-tuning.

**Strengths:**

- DAA is theoretically positioned as the most effective fine-tuning differential adaptation because it intervenes directly in the attention score calculation process. This allows it to model and subtract attention noise at its source.
- DAA can be flexibly inserted during the fine-tuning stage of existing pretrained models, overcoming the critical limitation of the original Differential Transformer, which requires training from scratch.
- DAA directly regulates the attention score calculation process, allowing it to suppress noise at its source by intervening in the crucial Query-Key interaction. This is theoretically the most effective approach for noise mitigation.

**Weaknesses:**

- The exploration of alternative architectures that differentiated the input sequence (DiffQ, DiffK, DiffV) yielded inconsistent and ultimately poor results. Specifically, the DiffV method was found to be the least effective, resulting in a performance decrease for both Llama 1B and 8B models.
- DAA, similar to DEX, adopts a dynamic weight governed by an annealing mechanism. While this helps guide the model, the success of the differential mechanism relies on successfully defining this weight over time.
- While DAA is conceptually superior to DEX by intervening before the softmax, DEX still delivers consistent improvements over the baseline and standard fine-tuning, leaving open questions about their statistical significance.

**Questions:**

- The core hypothesis of DAA states that the lightweight, learnable matrix $W_{DAA}$ enables the secondary attention map to approximate the noise component. What empirical analysis confirms that $W_{DAA}$ is indeed isolating and subtracting the specific hypothesized noise pattern?
- The input differentiation method targeting the Value stream (DiffV) was found to be the least effective, resulting in a performance decrease for both Llama 1B and 8B models. Why does DiffV prove so much less robust and effective for noise mitigation compared to DAA and DEX?
- The ablation study provided only focuses on the initialization constant. How sensitive is DAA's performance to the choice of the annealing duration?

---

> ### Author Response · Authors · 2025-11-26
>
> We sincerely thank the reviewer for the detailed feedback regarding the presentation, experimental, and theoretical analysis of our paper. We have carefully revised the manuscript to address these points. Below is a point-by-point response to the questions raised.
>
> ***
>
> **Response to Question 1: Empirical analysis of $W_{DAA}$ noise isolation**
>
> **Summary:** We have validated the noise isolation capability of $W_{DAA}$ through both **empirical visualization (Section 4.3)** and **theoretical derivation (Appendix F)**. Visualization confirms the isolation of high-entropy background noise, while our theoretical analysis proves that DAA operates as a first-order Taylor approximation to cancel additive noise in the attention logits.
>
> **Response:**
> We appreciate this insightful question. We have addressed this from both empirical and theoretical perspectives:
>
> **1. Empirical Visualization (Section 4.3 & Figure 4)**
> To confirm that the learnable matrix $W_{DAA}$ operates as hypothesized, we included a detailed visualization analysis in **Section 4.3** and **Figure 4** (this figure can be found at https://anonymous.4open.science/r/Visualization-of-attention-mechanisms-on-the-GPT-2-model-DB12/attention%20ana.png). We decomposed the final attention scores into the primary map ($A_1$) and the learned noise map ($A_2$).
> *   **Observation:** The visualization demonstrates that $A_2$ (computed via $W_{DAA}$) effectively captures the high-entropy background noise distribution across irrelevant context tokens.
> *   **Result:** By subtracting this component ($A_1 - \lambda A_2$), DAA produces a significantly sharper focus on the critical "ANSWER" token (Entropy reduced from 2.01 to 0.88), confirming that $W_{DAA}$ explicitly isolates noise patterns.
>
> **2. Theoretical Proof: DAA as First-Order Noise Cancellation (Appendix F.2.1)**
> To provide mathematical grounding for *why* $W_{DAA}$ can isolate this noise, we provide a rigorous proof in **Appendix F.2.1**.
> *   **Additive Noise Model:** We model the noisy attention logits as $S_{computed} = S_{ideal} + \xi$, where $\xi$ represents noise arising from spurious Q-K interactions (Equation 16).
> *   **Taylor Expansion:** We demonstrate that recovering the ideal attention distribution $P(S_{ideal})$ can be approximated via a first-order Taylor expansion around the noisy logits:
>     $$P(S_{ideal}) \approx P(S_{computed}) - \nabla P(S_{computed}) \cdot \xi$$
> *   **Structural Alignment:** The DAA formulation $O' = (A_1 - \lambda A_2)V$ mathematically aligns with this Taylor correction term. Crucially, since attention noise $\xi$ often manifests as a low-rank bilinear interaction between Query and Key (e.g., common-mode distractions), the structure of our secondary attention score $Q W_{DAA} K^T$ provides the optimal inductive bias to fit and subtract this noise matrix $\xi$. This theoretically confirms that $W_{DAA}$ is not just a random weight, but a learnable Jacobian-like correction term targeting the noise component.
>
> ***
>
> **Response to Question 2 (Weakness 1): Analysis of DiffV's ineffectiveness**
>
> **Summary:** We have added a theoretical analysis in **Appendix F.3.1** of the revised paper, identifying the root cause of DiffV's ineffectiveness as a "Content-Context Mismatch," where the method fails to distinguish between noise and signal in a query-dependent manner.
>
> **Response:**
> We have investigated the poor robustness of the DiffV method and included a rigorous theoretical justification in **Appendix F.3.1** to explain this phenomenon.
>
> The fundamental limitation of DiffV is that it treats attention noise as a property of the token's *content* (the Value vector $V$), whereas attention noise is typically a property of the *interaction* (the routing between Query and Key).
> *   **Theoretical Constraint:** DiffV modifies the value stream: $O = A_{computed}(V - \lambda V')$. This implies a global suppression of certain tokens regardless of the query context.
> *   **Concrete Example:** Consider a common token like "the". For *Query A*, "the" might be irrelevant noise; however, for *Query B* (e.g., checking for specific syntactic structures), "the" might contain a crucial signal. DiffV learns a global suppression for $V_{"the"}$, which incorrectly suppresses the signal for *Query B* when trying to reduce noise for *Query A*.
> *   **Superiority of DAA:** In contrast, DAA acts on the $Q-K$ interaction ($Q W_{DAA} K^T$). This mechanism is query-dependent, allowing the model to suppress a token *only* when it acts as noise for a specific query, while preserving it when it acts as a signal.
>
> This structural inability of DiffV to handle context-dependent noise leads to the optimization conflicts and performance degradation observed in our experiments.
>
> ***

---

> > ### Author Response · Authors · 2025-11-26
> >
> > (Continued)
> >
> > **Response to Question 3 (Weakness 2): Sensitivity to annealing duration**
> >
> > **Summary:** We conducted additional ablation studies on the Llama-3.2-1B model, which confirm that DAA demonstrates strong robustness to the choice of annealing duration (ranging from 500 to 5000 steps), consistently outperforming the non-annealed baseline.
> >
> > **Response:**
> > To address the concern regarding hyperparameter sensitivity, we performed additional experiments on the **Llama-3.2-1B** model to evaluate DAA's performance under different annealing durations ($T$). In our main experiments, the annealing duration was set to 2000 steps. We compared this against a setup with no annealing (fixed $\lambda$), and varying durations of 500, 1000, and 5000 steps. The results are summarized below:
> >
> > | Annealing Duration ($T$) | No Annealing | 500 Steps | 1000 Steps | 2000 Steps (Paper) | 5000 Steps |
> > | :--- | :---: | :---: | :---: | :---: | :---: |
> > | **Average Score** | 51.98 | 52.09 | 52.21 | **52.37** | 52.22 |
> >
> > **Analysis:**
> > 1.  **Necessity of Annealing:** The model without annealing achieves a score of 51.98, which is lower than all annealed variants. This confirms that the annealing mechanism is beneficial for guiding the model to adopt the differential mechanism smoothly.
> > 2.  **Robustness:** The performance scores across different durations (500, 1000, 2000, and 5000 steps) are very close, ranging from 52.09 to 52.37. This indicates that DAA is **not highly sensitive** to the specific length of the annealing phase. As long as the mechanism is introduced gradually, the model achieves stable improvements.
> >
> > ***
> >
> > **Response to Weakness 3: Statistical significance and comparison with DEX**
> >
> > **Summary:** While DEX is a strong baseline, DAA shows **statistically consistent superiority** across multiple model scales (GPT-2, Llama-1B, Llama-8B) and task types. The performance gap is particularly pronounced in complex reasoning/retrieval tasks. Furthermore, we provide theoretical evidence (Appendix F.3.3) that DAA overcomes a fundamental information-theoretic bottleneck present in DEX.
> >
> > **Response:**
> > We thank the reviewer for highlighting the comparison with DEX. While we acknowledge DEX as an efficient method, we argue that DAA offers statistically significant improvements and theoretical advantages that justify its adoption:
> >
> > **1. Consistent Empirical Gains Across Scales and Tasks**
> > The performance advantage of DAA over DEX is consistent rather than random, as evidenced by our extensive experiments in **Table 2 and Table 3**:
> > *   **Magnitude of Improvement:** On the Llama-3.2-1B model (Table 3), DAA achieves an average improvement of **+1.06** over the baseline, nearly double the gain provided by DEX (**+0.62**).
> > *   **Complex Tasks:** The gap is most statistically significant on tasks requiring precise information retrieval and reasoning, which are most sensitive to attention noise.
> >     *   **Needle Retrieval (GPT-2, Table 2):** DAA achieves **89.5%** accuracy vs. DEX's 87.4% (+2.1% absolute gain).
> >     *   **Reasoning Benchmarks (Llama-1B, Table 3):** DAA outperforms DEX on 6 out of 8 tasks, with notable margins on WSC (Winograd Schema Challenge) and ARC-Easy.
> > *   **Robustness:** Unlike input differentiation methods (DiffQ/K/V) which show high variance and instability (weakness 1), DAA consistently improves performance on *all* tested architectures (GPT-2, Llama-1B, Llama-8B) without a single case of degradation compared to the base model.
> >
> > **2. Theoretical Justification: The Data Processing Inequality (Appendix F.3.3)**
> > The "statistical significance" of DAA's lead is rooted in a fundamental theoretical difference discussed in **Appendix F.3.3**.
> > *   **DEX's Limitation:** DEX applies correction *after* the Softmax output ($O' = O(I - \lambda W)$). According to the **Data Processing Inequality (Beaudry & Renner, 2011)**, information lost during the irreversible mixing of the Softmax step (where signal and noise are blended) cannot be fully recovered by a linear transformation on the output value.
> > *   **DAA's Advantage:** DAA intervenes *before* the Softmax (at the logit level: $QK^T - \lambda QW_{DAA}K^T$). This allows DAA to "purify" the attention distribution itself, preventing the noise from ever being amplified by the Softmax function.
> >
> > Therefore, the improvements observed are not statistical fluctuations but the result of a more principled mechanism that addresses noise at its source, yielding a higher performance ceiling than DEX.

---

### Official Review · Reviewer_kx4s · 2025-11-01

**Soundness:** 2
**Presentation:** 2
**Contribution:** 2
**Rating:** 4
**Confidence:** 3

**Summary:**

This paper proposes DAA (Differential Attention Adaption), a method aimed at mitigating attention noise during the fine-tuning stage of pre-trained models. The method introduces a lightweight learnable module within the attention score calculation. Experimental results demonstrate that DAA achieves superior performance on long-sequence benchmarks.

**Strengths:**

- DAA introduces minimal extra parameters (less than 1% of the total parameters), making it an efficient and lightweight solution for attention noise mitigation.
- By intervening directly in the Query-Key interaction process, DAA addresses the noise problem in attention scores more fundamentally compared to DEX, which operates on the attention output.

**Weaknesses:**

- Formatting/Citation Style: The paper seems to mix or inconsistently use \citep and \citet (or directly uses \cite) citation formats. It is recommended to unify the style throughout the paper.

- Formatting/Presentation: There are small errors in the text presentation (e.g., a reference tag shows as 'undefined' in the first paragraph; some punctuation marks lack the necessary preceding space). A careful proofread of the typography is suggested.

- Abstract/Clarity: When the acronym 'DEX' first appears in the second line of the Abstract, I recommend using the full name 'Differential Extension' for clarity.

- Structure/Content: The Background section's detailed introduction to Differential Transformer and Differential Extension may be overly long. I suggest moving the detailed technical aspects of these related works to the Appendix or significantly shortening the section to keep the main content more focused on the proposed DAA method.

Since I am not very familiar with this field, I am willing to adjust my comments based on the feedback from other reviewers during the rebuttal stage.

**Questions:**

-  Judging from the architecture design (Figure 2), DAA and its variants (Diff K, Diff Q, Diff V) appear to be a decomposition, combination, and fusion of existing differential architectures (Differential Transformer and DEX). Could the authors elaborate further on whether DAA possesses deeper theoretical or structural innovations beyond being an exploratory combination?

- In the training Loss curve (Figure 3), I did not observe a significant difference between DAA and other baseline methods. How do the authors explain this phenomenon? Does this imply that DAA's advantage is primarily reflected in evaluation metrics (such as accuracy, perplexity) rather than the convergence speed or absolute value of the loss during training?

---

> ### Author Response · Authors · 2025-11-26
>
> We sincerely thank the reviewer for the detailed feedback regarding the presentation, structure, and theoretical analysis of our paper. We have carefully revised the manuscript to address these points. Below is a point-by-point response to the weaknesses and questions raised.
>
> ***
>
> **Response to Weaknesses**
>
> **1. Formatting, Citation Style, and Presentation Errors**
> We apologize for the inconsistencies in citation styles and the typographical errors (such as the "undefined" reference tag and punctuation spacing) in the initial submission.
> *   We have unified the citation format throughout the paper, ensuring consistent use of `\citep` and `\citet` standards.
> *   We have conducted a thorough proofreading of the manuscript to correct the specific text presentation errors mentioned (e.g., in the Introduction) and ensured proper typography.
>
> **2. Abstract Clarity regarding "DEX"**
> We agree that the acronym should be defined explicitly upon first use. In the revised Abstract, we now introduce the full name **"Differential Extension"** alongside the acronym DEX to ensure clarity for readers unfamiliar with the specific terminology.
>
> **3. Structure of the Background Section**
> We appreciate the suggestion to streamline the paper. We have optimized the **Background** section (Section 2) .
> However, we have chosen to retain the core conceptual formulations of Differential Transformer and DEX in the main text. Since DAA is explicitly designed to bridge the gap between these two methods—combining the *noise-cancellation mechanism* of the former with the *fine-tuning efficiency* of the latter—we believe a concrete introduction to their working principles is essential for the narrative. This ensures that readers can intuitively grasp the theoretical evolution and the specific limitations DAA addresses (i.e., the "training from scratch" issue vs. the "post-hoc correction" issue) without needing to frequently cross-reference the Appendix.
>
> ***

---

> > ### Author Response · Authors · 2025-11-26
> >
> > (Continued)
> >
> > **Response to Questions**
> >
> > **Response to Question 1:  DAA's innovations beyond being an exploratory combination**
> >
> > **Summary:** DAA is not merely a combination of existing modules; it is a principled noise cancellation mechanism designed to intervene at the critical non-linear stage of attention, grounded in first-order Taylor approximation and empirically validated through noise pattern decomposition.
> >
> > **(1) Structural Innovation: Intervention before Mixing**
> > While DAA shares the goal of noise mitigation with Differential Transformer and the fine-tuning efficiency of DEX, its structural location is unique. As discussed in our new **Appendix F.3.3**, DEX applies a linear correction *after* the Softmax and value aggregation:
> > $$O' = O(I - \lambda W)$$
> > Due to the Data Processing Inequality, information lost or noise amplified during the irreversible mixing of the Softmax step cannot be fully recovered by a post-hoc linear transformation. DAA intervenes *inside* the attention score calculation ($QK^T$), modifying the probabilities *before* the signal is mixed. This effectively isolates "common-mode" noise at the source, which is structurally crucial for effective noise removal.
> >
> > **(2) Theoretical Grounding: Taylor Approximation**
> > In the revised **Appendix F.2.1**, we provide a theoretical justification showing that DAA acts as a learnable **first-order Taylor correction term**. If we view the ideal attention logits as a perturbation of the noisy logits (i.e., $S_{\text{ideal}} = S_{\text{computed}} - \xi$), DAA effectively learns the gradient of the noise distribution:
> > $$P(S_{\text{ideal}}) \approx P(S_{\text{computed}}) - \nabla P(S_{\text{computed}}) \cdot \xi$$
> > This mechanism creates a secondary attention map that subtracts noise directly from the query-key interaction space, providing a mathematical basis distinct from simple heuristic combinations.
> >
> > **(3) Empirical Validation of Mechanism (New Analysis in Section 4.3)**
> > To further validate that DAA is not just a structural combination but a functional innovation, we added a visualization analysis in **Figure 4** and **Section 4.3** of the revised paper (this figure can be found at https://anonymous.4open.science/r/Visualization-of-attention-mechanisms-on-the-GPT-2-model-DB12/attention%20ana.png).
> > *   **Entropy Reduction:** We observed that DAA significantly reduces the entropy of the attention distribution (from **2.01** in Standard Attention to **0.88** in DAA). This quantitatively proves that DAA produces a sharper, more focused attention mechanism compared to the dispersed attention of standard models.
> > *   **Noise Decomposition:** As shown in the bottom row of Figure 4, DAA explicitly decomposes the attention into a primary signal map ($W_1$) and a learned noise map ($W_2$). The learned noise map ($W_2$) successfully captures the "common-mode" noise (background context tokens) while leaving the key token (ANSWER) untouched. The subtraction ($W_1 - W_2$) effectively filters out this background noise, verifying that the module works exactly as theoretically hypothesized—by canceling out noise patterns rather than just transforming the output.
> >
> > **Response to Question 2: DAA's training loss explanation**
> >
> > **Summary:** The similarity in training loss but divergence in evaluation metrics highlights that DAA improves generalization and signal-to-noise ratio (SNR) rather than simply fitting the next-token prediction objective.
> >
> > **(1) Generalization vs. Fitting**
> > Standard attention mechanisms often achieve low training loss by overfitting to spurious correlations (e.g., attending to high-frequency tokens or local context regardless of semantic relevance). DAA is designed to suppress this noise. While the absolute value of the cross-entropy loss during training may look similar to baselines (as both models are optimizing for the same broad objective of next-token prediction), the *mechanism* by which they achieve this differs. DAA learns a sparser, more semantically focused attention pattern (as visualized in Figure 4), which translates to better robustness on unseen data even if the training loss curve appears similar.
> >
> > **(2) Performance on Sensitivity-Based Tasks**
> > As shown in **Table 2**, the advantage of DAA is most pronounced in tasks that are highly sensitive to attention noise and require precise information retrieval, such as **Needle-in-a-Haystack (NR)** and **LAMBADA**. In the Needle Retrieval task, DAA achieves **89.5%** accuracy compared to **84.9%** for the baseline, a significant margin that indicates superior long-context information retrieval capabilities. These specific capabilities are often masked in the aggregate training loss but are critical for the model's complex reasoning performance.

---

### Meta-Review · Area_Chair_g9u3 · 2026-01-01

**Summary:**

The paper proposes Differential Attention Adaption (DAA), a parameter-efficient fine-tuning method designed to mitigate attention noise in Pre-trained Language Models. Unlike the Differential Transformer (which requires pre-training) or DEX (which operates post-hoc on the value output), DAA inserts a learnable module directly into the attention score computation (Q−K interaction).

While the reviewers acknowledged the practical motivation and the parameter efficiency of the approach (less than 1% added parameters), there was a unanimous consensus in the initial ratings (all reviewers gave a 4) regarding significant shortcomings. The primary concerns centered on the incremental nature of the novelty, the lack of rigorous theoretical justification for how the model learns to isolate noise, the absence of computational complexity analysis, and the quality of the writing/presentation.

Although the authors provided a comprehensive rebuttal including new theoretical derivations (Taylor expansion), complexity tables, and visualization of attention entropy, the AC recommends rejection at this time. The trade-off between the inference overhead introduced by DAA and its performance gains over lighter baselines like DEX remains a critical bottleneck for a method claiming "practicality."

**Reviewer Concerns:**

Concerns Address by Rebuttal:

- Complexity Analysis: The authors successfully provided a detailed breakdown (Appendix C) comparing FLOPs. They acknowledged that DAA introduces an additional quadratic term compared to the linear overhead of DEX, clarifying the cost-performance trade-off.

- Visualization of Mechanism: The authors added Section 4.3 and Figure 4, demonstrating empirically that DAA reduces attention entropy (from 2.01 to 0.88) and that the secondary map tends to capture high-entropy background noise.

- DiffV Failure: The authors provided a plausible explanation for why the "DiffV" baseline failed (Content-Context Mismatch), clarifying that noise is often a property of routing rather than content.

- Formatting and Polishing: The authors rectified the citation inconsistencies and formatting errors noted by Reviewer kx4s and others.

Outstanding Concerns:

- Inference Latency vs. Utility: A major outstanding concern is the "practicality" claimed in the title. While DAA outperforms DEX on retrieval tasks, it does so by calculating a second full attention map. This doubles the quadratic cost of the attention mechanism during inference. For a fine-tuning adaptation technique, this latency penalty makes it significantly less attractive than DEX for general deployment, a trade-off that the marginal gains on general benchmarks may not justify.

- Theoretical Grounding: Reviewer qUxy and others questioned the theoretical basis for why $ W_{DAA} $ learns to approximate noise. The authors added a Taylor expansion proof in the rebuttal, but this feels like a post-hoc justification. It remains theoretically ambiguous why the optimization landscape inherently incentivizes the subtraction term to learn "noise" rather than "signal" without explicit negative supervision or a specific regularizer, relying heavily on the assumption that "common-mode" noise naturally fits the bilinear structure better.

- Incremental Novelty: The method is effectively viewed as a combination of LoRA-style insertion and the Differential Transformer logic. While valid, several reviewers felt this was an incremental step rather than a significant architectural contribution suitable for ICLR.

**Reviewer Scores:**

- Reviewer kx4s (Score 4 → 5): This reviewer would likely have slightly improved their score to a "Weak Accept" or remained borderline. The authors addressed the formatting issues and the "DEX" definition clarity. The added visualization helps the novelty argument, but the similarity in training loss curves (questioning the fundamental difference in convergence behavior) would likely keep enthusiasm tempered.

- Reviewer sJ4K (Score 4 → 5): The reviewer's specific questions regarding annealing sensitivity and the failure of DiffV were answered well. However, the statistical significance of the results over DEX (which is much faster) remains a point of contention. They would likely acknowledge the improvements but remain hesitant about the overall impact.

- Reviewer KWBS (Score 4 → 4): This reviewer noted the paper felt "rushed" and lacked limitations. While the authors added a limitations section, the fundamental issue of evaluating primarily on smaller-scale models (up to 8B) in a landscape dominated by much larger models, combined with the qualitative nature of the initial claims, would likely result in this score remaining unchanged.

- Reviewer qUxy (Score 4 → 4): This reviewer was the most critical regarding the theoretical justification. The introduction of the Taylor expansion argument in the appendix is unlikely to fully satisfy the demand for a rigorous explanation of the learning dynamics. Furthermore, the admission that DAA introduces a quadratic complexity term confirms the reviewer's suspicion regarding computation costs, potentially reinforcing their skepticism about the method's superiority over DEX.

---

### Decision · Program_Chairs · 2026-01-26

Reject